# Preserving fairness and diagnostic accuracy in private large-scale AI models for medical imaging
Soroosh Tayebi Arasteh [1,6] ✉, Alexander Ziller [2,3,6] ✉, Christiane Kuhl[1], Marcus Makowski [2], Sven Nebelung [1], Rickmer Braren [2], Daniel Rueckert [3], Daniel Truhn [1,7] ✉ & Georgios Kaissis [2,3,4,5,7] ✉

## Abstract

**Background** Artificial intelligence (AI) models are increasingly used in the medical domain. However, as medical data is highly sensitive, special precautions to ensure its protection are required. The gold standard for privacy preservation is the introduction of differential privacy (DP) to model training. Prior work indicates that DP has negative implications on model accuracy and fairness, which are unacceptable in medicine and represent a main barrier to the widespread use of privacy-preserving techniques. In this work, we evaluated the effect of privacy-preserving training of AI models regarding accuracy and fairness compared to non-private training.

**Methods** We used two datasets: (1) A large dataset ($N = 193,311$) of high quality clinical chest radiographs, and (2) a dataset ($N = 1625$) of 3D abdominal computed tomography (CT) images, with the task of classifying the presence of pancreatic ductal adenocarcinoma (PDAC). Both were retrospectively collected and manually labeled by experienced radiologists. We then compared non-private deep convolutional neural networks (CNNs) and privacy-preserving (DP) models with respect to privacy-utility trade-offs measured as area under the receiver operating characteristic curve (AUROC), and privacy-fairness trade-offs, measured as Pearson's r or Statistical Parity Difference.

**Results** We find that, while the privacy-preserving training yields lower accuracy, it largely does not amplify discrimination against age, sex or co-morbidity. However, we find an indication that difficult diagnoses and subgroups suffer stronger performance hits in private training.

**Conclusions** Our study shows that – under the challenging realistic circumstances of a real-life clinical dataset – the privacy-preserving training of diagnostic deep learning models is possible with excellent diagnostic accuracy and fairness.

## Plain Language Summary

Artificial intelligence (AI), in which computers can learn to do tasks that normally require human intelligence, is particularly useful in medical imaging. However, AI should be used in a way that preserves patient privacy. We explored the balance between maintaining patient data privacy and AI performance in medical imaging. We use an approach called differential privacy to protect the privacy of patients' images. We show that, although training AI with differential privacy leads to a slight decrease in accuracy, it does not substantially increase bias against different age groups, genders, or patients with multiple health conditions. However, we notice that AI faces more challenges in accurately diagnosing complex cases and specific subgroups when trained under these privacy constraints. These findings highlight the importance of designing AI systems that are both privacy-conscious and capable of reliable diagnoses across patient groups.

The development of artificial intelligence (AI) systems for medical applications represents a delicate trade-off: On the one hand, diagnostic models must offer high accuracy and certainty, as well as treat different patient groups equitably and fairly. On the other hand, clinicians and researchers are subject to ethical and legal responsibilities towards the patients whose data is used for model training. In particular, when diagnostic models are published to third parties whose intentions are impossible to verify, care must be undertaken to ascertain that patient privacy is not compromised.

[1]Department of Diagnostic and Interventional Radiology, University Hospital RWTH Aachen, Aachen, Germany. [2]Institute of Diagnostic and Interventional Radiology, Technical University of Munich, Munich, Germany. [3]Artificial Intelligence in Healthcare and Medicine, Technical University of Munich, Munich, Germany. [4]Department of Computing, Imperial College London, London, United Kingdom. [5]Institute for Machine Learning in Biomedical Imaging, Helmholtz Munich, Neuherberg, Germany. [6]These authors contributed equally: Soroosh Tayebi Arasteh, Alexander Ziller.[7]These authors jointly supervised this work: Daniel Truhn, Georgios Kaissis. ✉e-mail: soroosh.arasteh@rwth-aachen.de; alex.ziller@tum.de; dtruhn@ukaachen.de; g.kaissis@tum.de

Privacy breaches can occur, e.g., through data reconstruction, attribute inference or membership inference attacks against the shared model[1]. Federated learning[2–4] has been proposed as a tool to address some of these problems. However, it has become evident that training data can be reverse-engineered from federated systems, rendering them just as vulnerable to the aforementioned attacks as centralized learning[5]. Thus, it is apparent that formal privacy preservation methods are required to protect the patients whose data is used to train diagnostic AI models. The gold standard in this regard is differential privacy (DP)[6].

Most, if not all, currently deployed machine learning models are trained without any formal privacy-preservation technique. It is especially crucial to employ such techniques in federated scenarios, where much more granular information about the training process can be extracted, or even the training process itself can be manipulated by a malicious participant[7,8]. Moreover, trained models can be attacked to extract training data through so-called model inversion attacks[9–11]. We also note that such attacks work better if the models have been trained on less data, which is especially concerning since even most FDA-approved AI algorithms have been trained on fewer than 1000 cases[12]. Creating a one-to-one correspondence between a successful attack and the resulting "privacy risk" requires a case-by-case consideration. The legal opinion (e.g., the GDPR) seems to have converged on the notion of singling out/ re-identification. Even from the aspect of newer legal frameworks, such as the EU AI act, which demand "risk moderation" rather than directly specifying "privacy requirements,", DP can be seen as the optimal tool as it can quantitatively bound both the risk of membership inference (MI)[13,14] and data reconstruction[15]. Moreover, this was also shown empirically for both aforementioned attack classes[16–19]. It is also known that DP, contrary to de-identification procedures such as $k$-anonymity, provably protects against the notion of singling out[20,21].

DP is a formal framework encompassing a collection of techniques to allow analysts to obtain insights from sensitive datasets while guaranteeing the protection of individual data points within them. DP thus is a property of a data processing system which states that the results of a computation over a sensitive dataset must be approximately identical whether or not any single individual was included or excluded from the dataset. Formally, a randomized algorithm (mechanism) $\mathcal{M} : \mathcal{X} \rightarrow \mathcal{Y}$ is said to satisfy $(\varepsilon, \delta)$-DP if, for all pairs of databases $D, D' \in \mathcal{X}$ which differ in one row and all $S \subseteq \mathcal{Y}$, the following holds:

$$\Pr\left(\mathcal{M}(D) \in S\right) \le e^{\varepsilon} \Pr\left(\mathcal{M}(D') \in S\right) + \delta, \qquad (1)$$

where the guarantee is given over the randomness of $\mathcal{M}$ and holds equally when $D$ and $D'$ are swapped. In more intuitive terms, DP is a guarantee given from a data processor to a data owner that the risks of adverse events which can occur due to the inclusion of their data in a database are bounded compared to the risks of such events when their data is not included. The parameters $\varepsilon$ and $\delta$ together form what is typically called a privacy budget. Higher values of $\varepsilon$ and $\delta$ correspond to a looser privacy guarantee and vice versa. With some terminological laxity, $\varepsilon$ can be considered a measure of the privacy loss incurred, whereas $\delta$ represents a (small) probability that this privacy loss is exceeded. For deep learning workflows, $\delta$ is set to around the inverse of the database size. We note that, although mechanisms exist where $\delta$ denotes a catastrophic privacy degradation probability, the sampled Gaussian mechanism used to train neural networks does not exhibit this behavior. The fact that quantitative privacy guarantees can be computed over many iterations (compositions) of complex algorithms like the ones used to train neural networks is unique to DP. This process is typically referred to as privacy accounting. Applied to neural network training, the randomization required by DP is ensured through the addition of calibrated Gaussian noise to the gradients of the loss function computed for each individual data point after they have been clipped in $\ell_2$-norm to ensure that their magnitude is bounded[22], where the clipping threshold is an additional hyperparameter in the training process.

DP does not only offer formal protection, but several works have also empirically shown the connection between the privacy budget and the success of membership inference[16] and data reconstruction attacks[17,19,23]. We note that absolute privacy (i.e., zero risk) is only possible if no information is present[24]. This is, for example, the case in encryption methods, which are perfectly private as long as data is not decrypted. Note that training models e.g., via homomorphic encryption does, however, not offer such perfect privacy guarantees, as the information learned by the model is actually revealed at inference time through the model's predictions. Thus, without the protection of differential privacy, no formal barrier stands between the sensitive data and an attacker (beyond potential imperfections of the attack algorithm, which are usually not controllable a priori). DP offers the ability to upper-bound the risk of successful privacy attacks while still being able to draw conclusions from the data. Determining the exact privacy budget is challenging, as it is a matter of policy. The technical perspective can provide insight into the appropriate budget level, as it is possible to quantify the risk of a successful attack at a given privacy budget compared to the model utility that can be achieved. The trade-offs between model utility and privacy preservation are also a matter of ethical, societal and political debate. The utilization of DP also creates two fundamental trade-offs: The first is a "privacy-utility trade-off," i.e., a reduction in diagnostic accuracy when stronger privacy guarantees are required[25,26]. The other trade-off is between privacy and fairness. Intuitively, the fact that AI models learn proportionally less about under-represented patient groups[27] in the training data is amplified by DP, leading to demographic disparity in the model's predictions or diagnoses[28]. Both of these trade-offs are delicate in sensitive applications, such as medical ones, as it is not acceptable to have wrong diagnoses or to discriminate against a certain patient group.

The need for the use of differential privacy (DP) has been illustrated by Packhäuser et al.[29], who showed that it is trivial to match chest x-rays of the same patient, which directly enables re-identification attacks; this was similarly shown in tabular databases by Narayanan et al.[30]. The training of deep neural networks on medical data with DP has so far not been widely investigated. Li et al.[31] investigated privacy-utility trade-offs in the combination of advanced federated learning schemes and DP methods on a brain tumor segmentation dataset. They find that DP introduces a considerable reduction in model accuracy in the given setting. Hatamizadeh et al.[23] illustrated that the use of federated learning alone can be unsafe in certain settings. Ziegler et al.[32] reported similar findings when evaluating privacy-utility trade-offs for a chest x-ray classification on a public dataset. These results also align with our previous work[17], where we demonstrated the utilization of a suite of privacy-preserving techniques for pneumonia classification in pediatric chest X-rays. However, the focus of this study was not to elucidate privacy-utility or privacy-fairness trade-offs, but to showcase that federated learning workflows can be used to train diagnostic AI models with good accuracy on decentralized data while minimizing data privacy and governance concerns. Moreover, we demonstrated that empirical data reconstruction attacks are thwarted by the utilization of differential privacy. In addition, the work did not consider differential diagnosis but only coarse-label classification into normal vs. bacterial or viral pneumonia.

In this work, we aim to elucidate the connection between using formal privacy techniques and the fairness towards underrepresented groups in the sensitive setting of medical use-cases. This is an important prerequisite for the deployment of ethical AI algorithms in such sensitive areas. However, so far, prior work is limited to benchmark computer vision datasets[33,34]. We thus contend that the widespread use of privacy-preserving machine learning requires testing under real-life circumstances. In the current study, we perform the first in-depth investigation into this topic. Concretely, we utilize a large clinical database of radiologist-labeled radiographic images, which has previously been used to train an expert-level diagnostic AI model, but otherwise not been curated or pre-processed for private training in any way. Furthermore, we analyze a dataset of abdominal 3D computed tomography (CT) images, where we classify the presence of a pancreatic ductal adenocarcinoma (PDAC). This mirrors the type of datasets available at clinical institutions. In this setting, we then study the extent of privacy-utility and privacy-fairness trade-offs in training advanced computer vision architectures.

To the best of our knowledge, our study is the first work to investigate the use of differential privacy in the training of complex diagnostic AI models on a real-world dataset of this magnitude (nearly 200,000 samples) and a 3D classification task, and to include an extensive evaluation of privacy-utility and privacy-fairness trade-offs.

Our results are of interest to medical practitioners, deep learning experts in the medical field and regulatory bodies such as legislative institutions, institutional review boards and data protection officers and we undertook specific care to formulate our main lines of investigation across the important axes delineated above, namely the provision of objective metrics of diagnostic accuracy, privacy protection and demographic fairness towards diverse patient subgroups.

Our main contributions can be summarized as follows: (1) We study the diagnostic accuracy ramifications of differentially private deep learning on two curated databases of medically relevant use-cases. We reach 97% of the non-private AUROC on the UKA-CXR dataset through the utilization of transfer learning on public datasets and careful choice of architecture. On the PDAC dataset, our private model at $\varepsilon = 8.0$ is not statistically significantly inferior compared to the non-private baseline. (2) We investigate the fairness implications of differentially private learning with respect to key demographic characteristics such as sex, age and co-morbidity. We find that – while differentially private learning has a mild fairness effect – it does not introduce significant discrimination concerns based on the subgroup representation compared to non-private training, especially at the intermediate privacy budgets typically used in large-scale applications.

## Methods
### Patient cohorts
We employed UKA-CXR[35,36], a large cohort of chest radiographs. The dataset consists of $N = 193,311$ frontal CXR images of 45,016 patients, all manually labeled by radiologists. The available labels include: pleural effusion, pneumonic infiltrates, and atelectasis, each separately for right and left lung, congestion, and cardiomegaly. The labeling system for cardiomegaly included five classes "normal," "uncertain," "borderline," "enlarged," and "massively enlarged." For the rest of the labels, five classes of "negative," "uncertain," "mild," "moderate," and "severe" were used. Data were split into $N = 153,502$ training and $N = 39,809$ test images using patient-wise stratification, but otherwise completely random allocation[35,36]. There was no

overlap between the training and test sets. Supplementary Table 1 shows the statistics of the dataset, which are further visualized in Supplementary Figs. 1 and 2.

In addition, we used an in-house dataset at Klinikum Rechts der Isar of 1625 abdominal CT scans from unique, consecutive patients, of which 867 suffered from pancreatic ductal adenocarcinoma (PDAC) (positive) and 758 were a control group without a tumor (negative). We split the dataset into 975 train and 325 validation and test images respectively. During splitting we maintained the ratio of positive and negative samples in all subsets.

The experiments were performed in accordance with relevant national and international guidelines and regulations. Approval for the UKA-CXR dataset by the Ethical Committee of the Medical Faculty of RWTH Aachen University has been granted for this retrospective study (Reference No. EK 028/19). Analogously, for the PDAC dataset, the protocol was approved by the Ethics Committee of Klinikum Rechts der Isar (Protocol Number 180/17S). Both institutional review boards did not require informed consent from subjects and/or their legal guardian(s) as this was a retrospective study. The study was conducted in accordance with the Declaration of Helsinki.

### Data pre-processing
We resized all images of the UKA-CXR dataset to $(512 \times 512)$ pixels. Afterward, a normalization scheme as described previously by Johnson et al.[37] was utilized by subtracting the lowest value in the image, dividing by the highest value in the shifted image, truncating values, and converting the result to an unsigned integer, i.e., in the range of [0,255]. Finally, we performed histogram equalization by shifting pixel values towards 0 or towards 255 such that all pixel values 0 through 255 have approximately equal frequencies[37].

We selected a binary classification paradigm for each label. The "negative" and "uncertain" classes ("normal" and "uncertain" for cardiomegaly) were treated as negative, while the "mild," "moderate," and "severe" classes ("borderline," "enlarged," and "massively enlarged" for cardiomegaly) were treated as positive.

For the PDAC dataset, we clipped the voxel density values of all CT scans to an abdominal window from −150 to 250 Hounsfield units and resized to a shape of $224 \times 224 \times 128$ voxels.

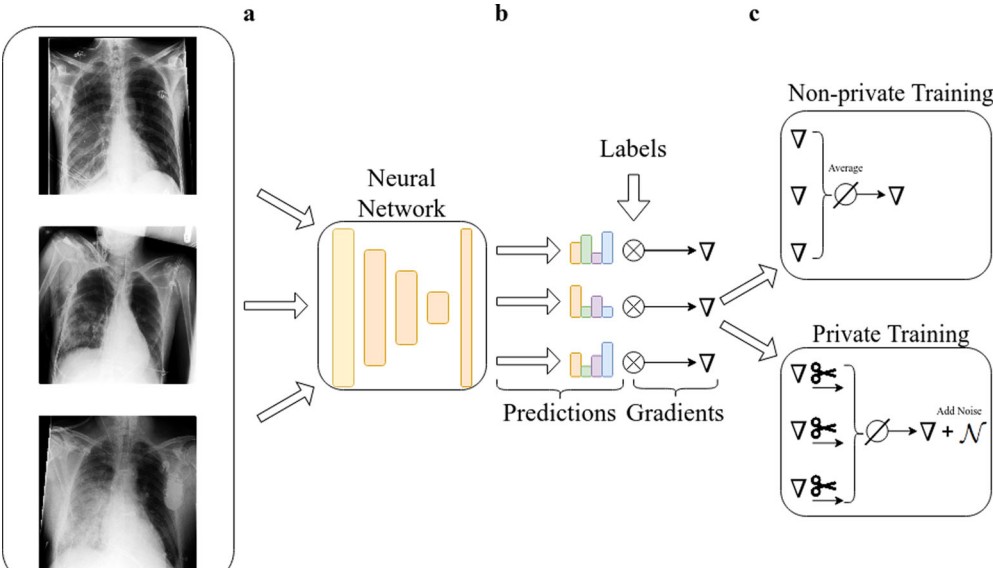

**Fig. 1 | Differences between the private and non-private training process of a neural network. a** Images from a dataset are fed to a neural network and predictions are made. **b** From the predictions and the ground truth labels, the gradient is calculated via backpropagation. (**(c)**, upper panel) In normal training all gradients are

averaged and an update step is performed. (**(c)**, lower panel) In private training, each per-sample gradient is clipped to a predetermined $\ell_2$-norm, averaged and noise proportional to the norm is added. This ensures that the information about each sample is upper-bounded and perturbed with sufficient noise.

**Table 1 | Summary of dataset statistics and results**

**UKA-CXR**

| Section | Label | Total μ | Total σ | Male μ | Male σ | Female μ | Female σ | [0,30] μ | [0,30] σ | [30,60] μ | [30,60] σ | [60,70] μ | [60,70] σ | [70,80] μ | [70,80] σ | [80,100] μ | [80,100] σ |
|---|---|---|---|---|---|---|---|---|---|---|---|---|---|---|---|---|---|
| Train | N | 153,502 | | 100,659 | | 52,843 | | 4279 | | 42,340 | | 36,882 | | 48,864 | | 21,137 | |
| Test | N | 39,809 | | 25,360 | | 14,449 | | 1165 | | 10,291 | | 10,025 | | 12,958 | | 5370 | |
| | Cardiomegaly | 18,616 | | 12,868 | | 5748 | | 334 | | 3853 | | 4714 | | 6837 | | 2876 | |
| | Congestion | 3275 | | 2206 | | 1069 | | 50 | | 817 | | 906 | | 991 | | 510 | |
| | Pl. Eff. R. | 3275 | | 2090 | | 1185 | | 52 | | 709 | | 847 | | 1248 | | 419 | |
| | Pl. Eff. L. | 2602 | | 1636 | | 966 | | 70 | | 589 | | 632 | | 894 | | 417 | |
| | Pn. Inf. R. | 4847 | | 3374 | | 1473 | | 184 | | 1322 | | 1367 | | 1361 | | 612 | |
| | Pn. Inf. L. | 3562 | | 2381 | | 1181 | | 143 | | 1087 | | 949 | | 959 | | 423 | |
| | Atel. R. | 3920 | | 2571 | | 1349 | | 127 | | 1010 | | 1056 | | 1272 | | 454 | |
| | Atel. L. | 3166 | | 2010 | | 1156 | | 119 | | 867 | | 774 | | 961 | | 444 | |
| AUROC | ε = 0.29 | 83.13 | 3.9 | 82.66 | 3.9 | 83.85 | 4.0 | 86.47 | 3.5 | 85.21 | 3.9 | 83.03 | 3.6 | 81.66 | 4.5 | 81.27 | 4.3 |
| | 0.54 | 84.00 | 3.8 | 83.61 | 3.8 | 84.61 | 3.9 | 86.43 | 3.1 | 85.96 | 3.7 | 83.96 | 3.5 | 82.69 | 4.5 | 82.15 | 4.2 |
| | 1.06 | 84.98 | 3.9 | 84.69 | 3.9 | 85.40 | 4.0 | 87.69 | 3.2 | 86.90 | 3.7 | 84.95 | 3.8 | 83.70 | 4.5 | 82.96 | 4.3 |
| | 2.04 | 85.80 | 3.9 | 85.52 | 3.9 | 86.19 | 3.9 | 88.77 | 3.3 | 87.53 | 3.8 | 85.88 | 3.8 | 84.47 | 4.4 | 83.85 | 4.3 |
| | 4.71 | 86.93 | 4.0 | 86.73 | 4.1 | 87.19 | 4.0 | 89.11 | 3.3 | 88.59 | 3.9 | 86.80 | 3.7 | 85.89 | 4.7 | 85.08 | 4.6 |
| | 7.89 | 87.36 | 4.1 | 87.12 | 4.2 | 87.66 | 4.1 | 89.72 | 4.1 | 88.97 | 3.9 | 87.26 | 3.9 | 86.30 | 4.7 | 85.48 | 4.8 |
| | ∞ | 89.71 | 3.8 | 89.46 | 3.9 | 90.06 | 3.8 | 91.64 | 3.5 | 90.99 | 3.4 | 89.73 | 3.8 | 88.73 | 4.4 | 88.18 | 4.5 |
| PtD | ε = 0.29 | | | −1.40 | 0.22 | +1.40 | 0.22 | +7.05 | 0.18 | −0.98 | 0.73 | +0.97 | 1.75 | −1.73 | 0.36 | −1.63 | 1.00 |
| | 0.54 | | | −1.56 | 0.10 | +1.56 | 0.10 | +7.20 | 0.21 | −0.80 | 0.52 | +1.95 | 0.48 | −2.65 | 0.31 | −1.23 | 0.56 |
| | 1.06 | | | −0.87 | 0.73 | +0.87 | 0.73 | +7.35 | 0.51 | −2.56 | 0.67 | +0.49 | 0.23 | −1.92 | 0.78 | −3.12 | 0.13 |
| | 2.04 | | | +0.15 | 0.42 | −0.15 | 0.42 | +6.12 | 0.92 | −1.80 | 0.39 | +1.50 | 0.00 | −2.80 | 0.30 | −1.61 | 0.15 |
| | 4.71 | | | −1.63 | 0.31 | +1.63 | 0.31 | +4.37 | 0.18 | −2.15 | 0.70 | +1.26 | 1.38 | −2.27 | 0.50 | −2.36 | 2.38 |
| | 7.89 | | | −0.66 | 0.75 | +0.66 | 0.75 | +5.53 | 0.92 | −1.27 | 0.04 | +1.21 | 0.22 | −1.33 | 0.06 | −2.89 | 0.52 |
| | ∞ | | | −0.34 | 0.47 | +0.34 | 0.47 | +4.00 | 0.60 | −1.32 | 0.65 | +0.21 | 0.66 | −0.43 | 0.95 | −2.67 | 0.20 |

**PDAC**

| Section | Label | Total μ | Total σ | Male μ | Male σ | Female μ | Female σ | Youngest 25% μ | Youngest 25% σ | Second 25% μ | Second 25% σ | Third 25% μ | Third 25% σ | Oldest 25% μ | Oldest 25% σ |
|---|---|---|---|---|---|---|---|---|---|---|---|---|---|---|---|
| Train | N | 975 | | 552 | | 423 | | 231 | | 290 | | 228 | | 226 | |
| Test | N | 325 | | 197 | | 127 | | 86 | | 85 | | 79 | | 75 | |
| | Tumor | 173 | | 95 | | 77 | | 23 | | 48 | | 54 | | 48 | |
| | Control | 152 | | 102 | | 50 | | 63 | | 37 | | 25 | | 27 | |
| AUROC | ε = 0.29 | 86.84 | 4.0 | 88.11 | 4.6 | 85.47 | 2.5 | 87.92 | 9.1 | 85.87 | 3.0 | 84.44 | 1.2 | 89.15 | 7.2 |
| | 0.54 | 92.60 | 1.3 | 93.62 | 1.5 | 91.00 | 0.9 | 93.77 | 3.2 | 91.97 | 1.2 | 90.05 | 0.4 | 95.63 | 2.3 |
| | 1.06 | 95.58 | 0.9 | 96.70 | 0.9 | 93.52 | 1.3 | 96.57 | 1.6 | 94.84 | 1.3 | 93.83 | 1.1 | 98.43 | 0.9 |
| | 2.04 | 97.49 | 0.4 | 98.50 | 0.3 | 95.36 | 0.9 | 97.98 | 0.9 | 96.90 | 0.8 | 97.06 | 0.9 | 99.36 | 0.6 |
| | 4.71 | 98.31 | 0.2 | 99.19 | 0.1 | 96.38 | 0.7 | 98.48 | 0.3 | 97.84 | 0.2 | 98.30 | 0.4 | 99.97 | 0.0 |

**Table 1 (continued) | Summary of dataset statistics and results**

| ε | μ | σ | μ | σ | μ | σ | μ | σ | μ | σ | μ | σ | μ | σ |
|---|---|---|---|---|---|---|---|---|---|---|---|---|---|---|
| 5.0 | 98.33 | 0.2 | 99.20 | 0.1 | 96.41 | 0.7 | 98.48 | 0.4 | 97.86 | 0.1 | 98.37 | 0.4 | 100.00 | 0.0 |
| 6.0 | 98.39 | 0.3 | 99.22 | 0.1 | 96.55 | 0.8 | 98.57 | 0.3 | 97.84 | 0.2 | 98.35 | 0.5 | 100.00 | 0.0 |
| 7.0 | 98.41 | 0.3 | 99.22 | 0.1 | 96.60 | 0.8 | 98.62 | 0.3 | 97.88 | 0.1 | 98.25 | 0.5 | 100.00 | 0.0 |
| 8.0 | 99.28 | 0.7 | 99.77 | 0.3 | 98.13 | 1.6 | 99.59 | 0.7 | 99.23 | 1.2 | 98.37 | 0.9 | 100.00 | 0.0 |
| ∞ | 99.70 | 0.2 | 99.97 | 0.1 | 99.01 | 0.6 | 99.98 | 0.0 | 99.94 | 0.1 | 98.47 | 0.9 | 100.00 | 0.0 |
| **PtD** 0.29 | | | +3.27 | 5.38 | −3.27 | 5.38 | +9.03 | 1.32 | +1.87 | 2.12 | −9.54 | 4.33 | −2.04 | 4.74 |
| 0.54 | | | +1.02 | 0.76 | −1.02 | 0.76 | +3.17 | 0.54 | +0.34 | 1.44 | −7.02 | 4.39 | +3.42 | 3.28 |
| 1.06 | | | +1.29 | 1.27 | −1.29 | 1.27 | −0.18 | 3.85 | +0.20 | 1.66 | −3.58 | 3.53 | +3.69 | 2.83 |
| 2.04 | | | +3.00 | 0.78 | −3.00 | 0.78 | −1.97 | 0.65 | −3.16 | 2.47 | +1.55 | 0.62 | +4.00 | 3.47 |
| 4.71 | | | +4.58 | 1.33 | −4.58 | 1.33 | −3.29 | 1.23 | −2.34 | 1.20 | +1.47 | 1.46 | +4.62 | 1.46 |
| 5.0 | | | +4.85 | 1.37 | −4.85 | 1.37 | −2.62 | 0.82 | −2.73 | 1.16 | +1.61 | 0.87 | +4.18 | 1.90 |
| 6.0 | | | +4.41 | 0.53 | −4.41 | 0.53 | −2.10 | 2.06 | −2.20 | 0.64 | +1.05 | 2.10 | +3.60 | 1.14 |
| 7.0 | | | +3.19 | 1.27 | −3.19 | 1.27 | −1.99 | 2.97 | −3.68 | 1.21 | +1.20 | 2.31 | +4.93 | 2.02 |
| 8.0 | | | +3.28 | 2.61 | −3.28 | 2.61 | −2.45 | 1.51 | −1.45 | 2.28 | +1.58 | 2.44 | +2.62 | 1.61 |
| ∞ | | | +2.81 | 2.38 | −2.81 | 2.38 | −1.21 | 1.59 | −0.18 | 1.16 | −0.33 | 1.71 | +1.87 | 1.22 |

Diagnostic performance of patient subgroups for the UKA-CXR and PDAC datasets. We report the number of cases over subgroups and labels. All values refer to the test set. Total denotes the results on the entire test set. AUROC denotes the area under the receiver operating characteristic curve. PtD is the statistical parity difference of each subgroup. PDAC stands for presence of pancreatic ductal adenocarcinoma. μ are mean values, σ shows the standard deviation calculated over 1000 bootstrapping samples (UKA-CXR) respectively 3 independent model trainings (PDAC). All results are in percent.

## Deep learning process

**Network architecture.** For both datasets, we employed the ResNet9 architecture introduced in ref. 38 as our classification architecture. For the UKA-CXR dataset, images were expanded to $(512 \times 512 \times 3)$ for compatibility with the neural network architecture. The final linear layer reduces the $(512 \times 1)$ output feature vectors to the desired number of diseases to be predicted, i.e., 8. The sigmoid function was utilized to convert the output predictions to individual class probabilities. The full network contained a total of 4.9 million trainable parameters. For the PDAC dataset, we used the conversion proposed by Yang et al.[39] to convert the model to be applicable to 3D data, which in brief applies 2D-convolutional filters along axial, coronal, and sagittal axes separately. Our utilized ResNet9 network employs the modifications proposed by Klause et al.[38] and by He et al.[40]. Batch Normalization[41] is incompatible with DP-SGD, as per-sample gradients are required, and batch normalization inherently intermixes information of all images in one batch. Hence, we used group normalization[42] layers instead with 32 groups to be compatible with DP processing. For the CXR dataset we pretrained the network on the MIMIC Chest X-ray JPG dataset v2.0.0 (MIMIC-CXR),[43] consisting of $N = 210{,}652$ frontal images. All training hyperparameters were selected empirically based on their validation accuracy, while no systematic/automated hyperparameter tuning was conducted.

**Non-DP training.** For the UKA-CXR dataset, the Rectified Linear Unit (ReLU)[44,45] was chosen as the activation function in all layers. We performed data augmentation during training by applying random rotation in the range of $[-10, 10]$ degrees and medio-lateral flipping with a probability of 0.50. The model was optimized using the NAdam[46] optimizer with a learning rate of $5 \cdot 10^{-5}$. The binary weighted cross-entropy with inverted class frequencies of the training data was selected as the loss function. The training batch size was chosen to be 128. In the PDAC dataset, we used an unweighted binary cross-entropy loss as well as the NAdam optimizer with a learning rate of $2 \cdot 10^{-4}$.

**DP training.** For UKA-CXR, we chose Mish[47] as the activation function in all layers. No data augmentation was performed during DP training as we found further data augmentation during training to be harmful to accuracy. All models were optimized using the NAdam[46] optimizer with a learning rate of $5 \cdot 10^{-4}$. The binary weighted cross-entropy with inverted class frequencies of the training data was selected as the loss function. The maximum allowed gradient norm (see Fig. 1) was chosen to be 1.5 and the network was trained for 150 epochs for each chosen privacy budget. Each point in the batch was sampled with a probability of $8 \cdot 10^{-4}$ (128 divided by $N = 153{,}502$). For the PDAC dataset, we chose a clipping norm of 1.0, $\delta = 0.001$ and a sampling rate of 0.31 (512/1 625). In both cases, the noise multiplier was calculated such that for a given number of training steps, sampling rate, and maximum gradient norm the privacy budget was reached on the last training step. For the UKA-CXR dataset, the indicated privacy guarantees are "per record" since some patients have more than one image, while for the PDAC datasets, they are "per individual."

## Quantitative evaluation and statistical analysis

The area under the receiver operating characteristic curve (AUROC) was utilized as the primary evaluation metric. We report the average AUROC over all the labels for each experiment. The individual AUROC as well as all other evaluation metrics of individual labels are reported in the supplementary information (Supplementary Tables 2–8). For the UKA-CXR test set, we used bootstrapping with 1000 redraws for each measure to determine the statistical spread[48]. For calculating sensitivity, specificity, and accuracy, a threshold was chosen according to Youden's criterion[49], i.e., the threshold that maximized (true positive rate – false positive rate).

To evaluate the correlation between results of data subsets and their sample size, Pearson's r coefficient was used. To analyze fairness between

subgroups, the statistical parity difference[50] was used which is defined as

$$P(\hat{Y} = 1 | C = \text{Minority}) - P(\hat{Y} = 1 | C = \text{Majority}) \quad (2)$$

where $\hat{Y} = 1$ represents correct model predictions and $C$ is the group in question. Intuitively, it is the difference in classification accuracy between the minority and majority class and thus is optimally zero. Values larger than zero mean that there is a benefit for the minority class, while values smaller than zero mean that the minority class is discriminated against.

### Reporting summary
Further information on research design is available in the Nature Portfolio Reporting Summary linked to this article.

## Results
### High classification accuracy is attainable despite stringent privacy guarantees
Table 1 shows an overview of our results for all subgroups. Supplementary Tables 2–8 show the per-diagnosis evaluation results for non-DP and DP training for different $\varepsilon$ values. On the UKA-CXR dataset our non-private model achieves an AUROC of 89.71% over all diagnoses. It performs best on pneumonic infiltration on the right (AUROC=94%) while struggling the most to accurately classify cardiomegaly (AUROC=84%). Training with DP decreases all results slightly yet significantly (Hanley & McNeil-test $p$-value < 0.001, 1 000 bootstrapping redraws) and achieves an overall AUROC of 87.36%. The per-diagnosis performance ranges from 92% (pleural effusion right) to 81% AUROC (congestion). We next consider classification performance at a very strong level of privacy protection (i.e., at $\varepsilon < 1$). Here, at an $\varepsilon$-budget of only 0.29, our model achieves an average AUROC of 83.13% over

all diagnoses. A visual overview is displayed in Fig. 2, which shows the average AUROC, accuracy, sensitivity, and specificity values over all labels.

On the PDAC dataset, we found that, while non-private training achieved almost perfect results on the test set the loss in utility for private training at $\varepsilon = 8$ is statistically non-significant (Hanley & McNeil-test $p$-value: 0.34, 3 independent experiments) compared to non-private training. Again, with lower privacy budgets, model utility decreases, but even at a very low privacy budget of $\varepsilon = 1.06$, we observe an average AUROC score of 95.58%.

Moreover, for UKA-CXR, the use of pre-training helps to boost model performance and reduce the amount of additional information the model needs to learn "from scratch" and consequently reduces the privacy budgets required (refer to Supplementary Fig. 3). This appears to primarily benefit the under-represented groups in the dataset. Conversely, non-private training, whether initialized with pre-training weights or trained from scratch, tends to yield comparable diagnostic results, as the latter network can leverage a greater amount of information. These findings are in line with the observations on the PDAC dataset (where no pretrained weights were available), namely that, at low privacy budgets, specific patient groups suffer a higher discrimination.

For the purpose of further generalization, we replicated the experiments using three other network architectures. All three models displayed a trend consistent with the utility penalties we observed for ResNet9 in both DP and non-DP training (see Supplementary Fig. 4). For further details, we refer to the supplementary information.

### Diagnostic accuracy is correlated with patient age and sample size for both private and non-private models
Fig. 3 shows the difference in classification performance on the UKA-CXR dataset for each diagnosis between the non-private model evaluation and its

**a**
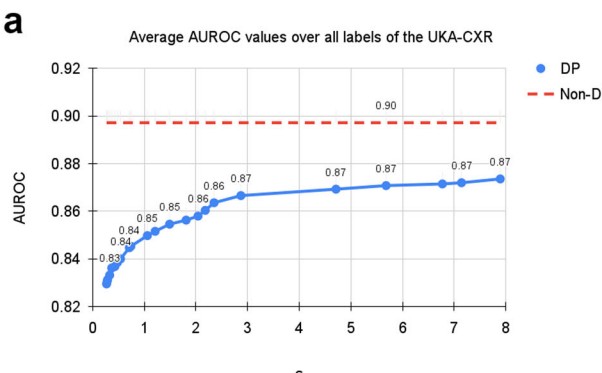

**b**
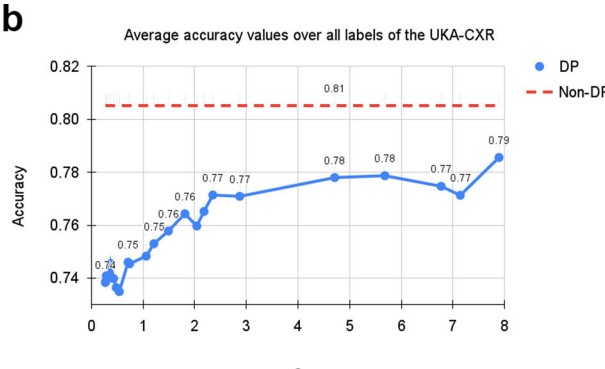

**c**
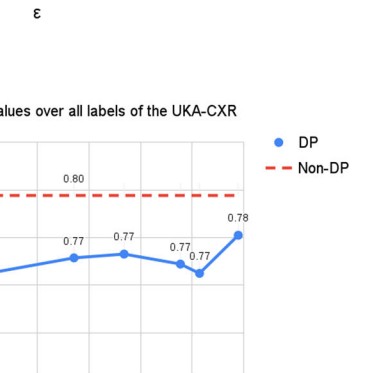

**d**
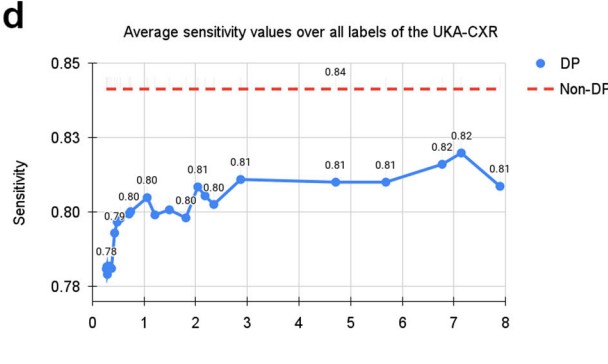

**Fig. 2 | Average results of training with differential privacy (DP) with different $\varepsilon$ values for $\delta = 6 \cdot 10^{-6}$.** The curves show the average (**a**) area under the receiver operating characteristic curve (AUROC), (**b**) accuracy, (**c**) specificity, and (**d**) sensitivity values over all labels, including cardiomegaly, congestion, pleural effusion right, pleural effusion left, pneumonic infiltration right, pneumonic infiltration left, atelectasis right, and atelectasis left tested on $N = 39,809$ test images. The training

dataset includes $N = 153,502$ images. Note, that the AUROC is monotonically increasing, while sensitivity, specificity and accuracy exhibit more variation. This is due to the fact that all training processes were optimized for the AUROC. Dashed lines correspond to the non-private training results. Source data are provided as a Source Data file.

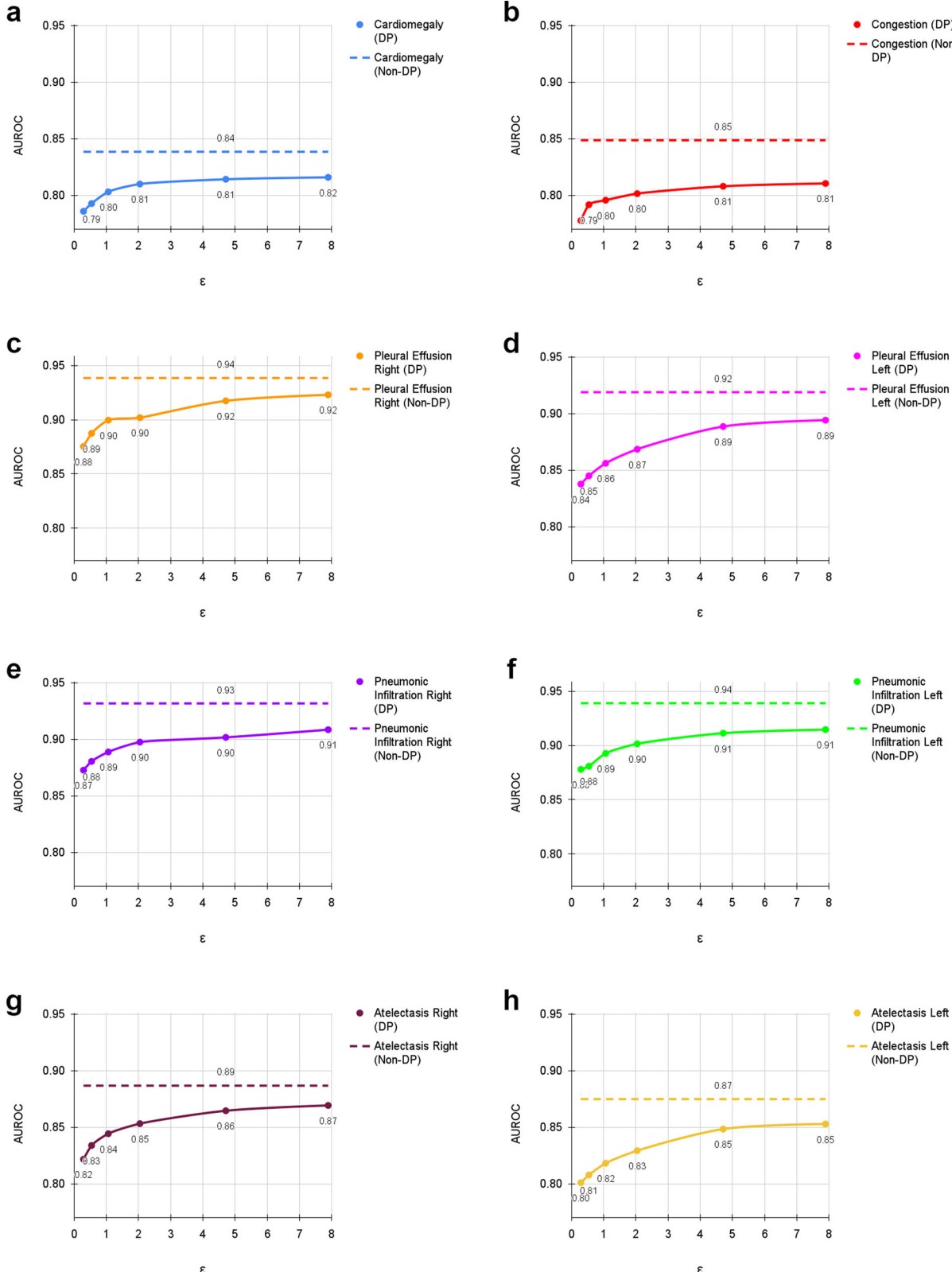

**Fig. 3 | Evaluation results of training with differential privacy (DP) and without DP with different $\epsilon$ values for $\delta = 6 \cdot 10^{-6}$.** The results show the individual area under the receiver operating characteristic curve (AUROC) values for (**a**) cardiomegaly, (**b**) congestion, (**c**) pleural effusion right, (**d**) pleural effusion left, (**e**) pneumonic infiltration right, (**f**) pneumonic infiltration left, (**g**) atelectasis right, and (**h**) atelectasis left tested on $N = 39,809$ test images. The training dataset includes $N = 153,502$ images. Dashed lines correspond to the non-private training results. Source data are provided as a Source Data file.

private counterpart compared to the sample size (that is, the number of available samples with a given label) within our dataset. At an $\epsilon$ = 7.89, the largest difference of AUROC between the non-private and privacy-preserving model was observed for congestion (3.82%) and the smallest difference was observed for pleural effusion right (1.55%, see Fig. 3). Of note, there is a visible trend (Pearson's r: 0.44) whereby classes in which the model exhibits good diagnostic performance in the non-private setting also suffer the smallest drop in the private setting. On the other hand, classes that are already difficult to predict in the non-private case deteriorate the most in terms of classification performance with DP (see Supplementary Fig. 9). Both non-private (Pearson's r: 0.57) and private (Pearson's r: 0.52) diagnostic AUROC exhibit a weak correlation with the number of samples available for each class (see Supplementary Fig. 9). However, the drop in AUROC between private and non-private training is not correlated with the sample size (Pearson's r: 0.06). On the PDAC dataset, patients with a tumor are overrepresented and in the non-private case diagnosed more accurately. Not surprisingly, the classification performance is thus also higher for private trainings except for the most restrictive privacy budget (see Supplementary Figs. 5–8).

Furthermore, we evaluated our models based on age range and patient sex (Table 1 and Figs. 4 and 5). Additionally, we calculated statistical parity difference for those groups to obtain a measure of fairness (Table 1). On the UKA-CXR dataset all models performed the best on patients younger than 30 years of age. It appears that, the older patients are, the greater the difficulty for the models to predict the labels accurately. Statistical parity difference

scores are slightly negative for the age groups between 70 and 80 years and older than 80 years for all models, indicating that the models discriminate slightly against these groups. In addition, while for the aforementioned age groups the discrimination does not change with privacy levels, younger patients become more privileged as privacy increases. This finding indicates that – for models which are most protective of data privacy – young patients benefit the most, despite the group of younger patients being smaller overall. For patient sex, models show slightly better performance for female patients and slightly discriminate against male patients (Table 1). Statistical parity does not appear to correlate (Pearson's r: 0.13) with privacy levels.

On the PDAC dataset, we observed that, for all levels of privacy including non-private training, classification performance was worse for female patients compared to male patients, who are over-represented in the dataset. However, there is no trend observable between the privacy level and the parity difference. When analysing results of subgroups separated by patient age, we observed similarly to UKA-CXR that in all settings, statistical parity differences are on average better for younger patients compared to older ones. Just as in the UKA-CXR dataset, we found that the more restrictive the privacy budget is set, the stronger the privilege enjoyed by younger patients. We furthermore observed that the control group (i.e., no tumor) has an over-representation of both male patients and young patients, which consequently both exhibit better performance compared to the rest of the cohort. Conversely, female patients as well as older patients, have a higher chance of misclassification and are more abundant in the tumor group.

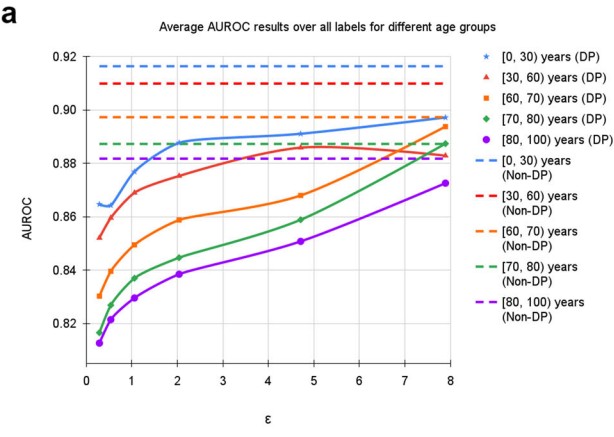

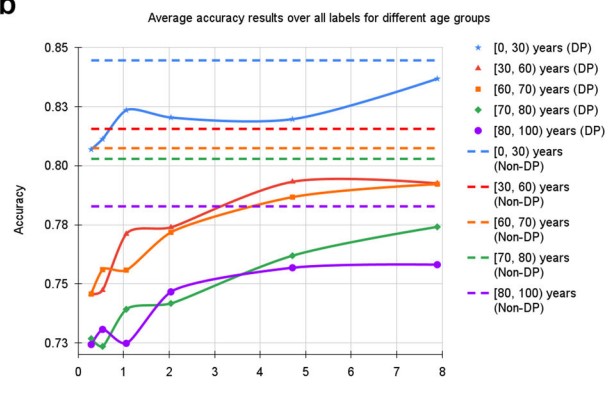

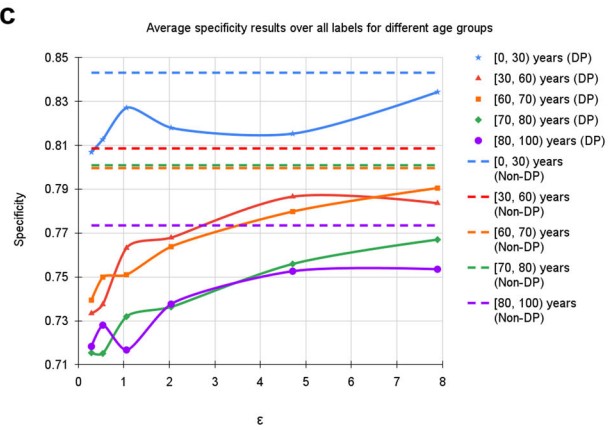

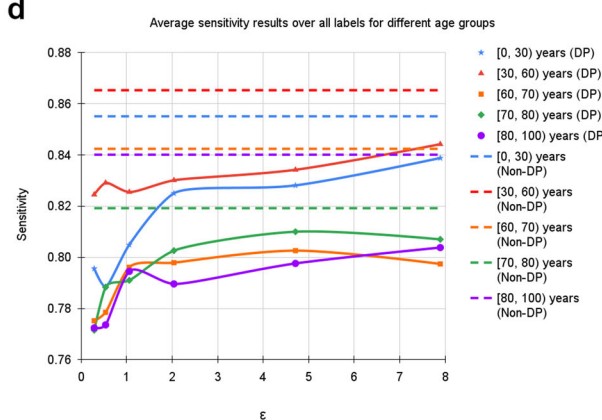

**Fig. 4 | Average results of training with differential privacy (DP) with different $\epsilon$ values for $\delta = 6 \cdot 10^{-6}$, separately for samples of different age groups including [0, 30), [30, 60), [60, 70), [70, 80), and [80, 100) years.** The curves show the average (**a**) area under the receiver operating characteristic curve (AUROC), (**b**) accuracy, (**c**) specificity, and (**d**) sensitivity values over all labels, including cardiomegaly, congestion, pleural effusion right, pleural effusion left, pneumonic infiltration right, pneumonic infiltration left, atelectasis right, and atelectasis left tested on $N$ = 39,809 test images. The training dataset includes $N$ = 153,502 images. Dashed lines in corresponding colors correspond to the non-private training results. Source data are provided as a Source Data file.

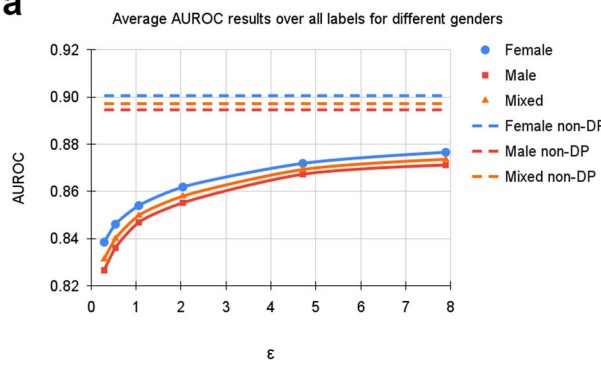

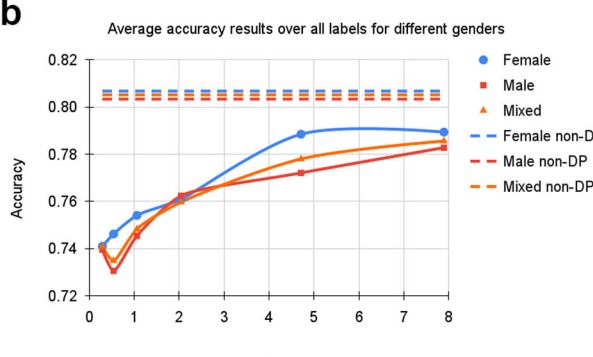

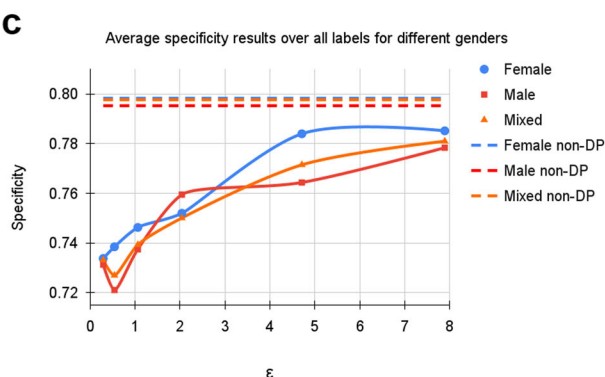

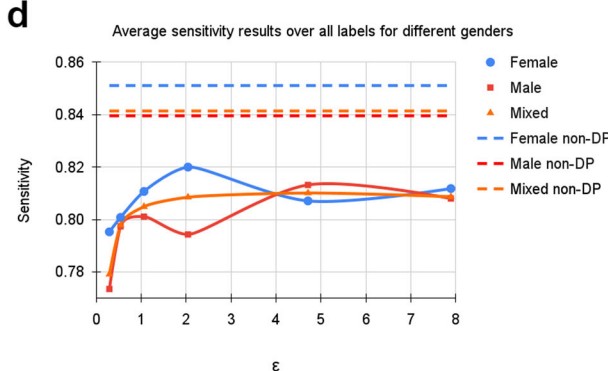

**Fig. 5 | Average results of training with differential privacy (DP) with different $\epsilon$ values for $\delta = 6 \cdot 10^{-6}$, separately for female and male samples.** The curves show the average (**a**) area under the receiver operating characteristic curve (AUROC), (**b**) accuracy, (**c**) specificity, and (**d**) sensitivity values over all labels, including cardiomegaly, congestion, pleural effusion right, pleural effusion left, pneumonic infiltration right, pneumonic infiltration left, atelectasis right, and atelectasis left tested on $N = 39,809$ test images. The training dataset includes $N = 153,502$ images. Note, that the AUROC is monotonically increasing, while sensitivity, specificity and accuracy exhibit more variation. This is due to the fact that all training processes were optimized for the AUROC. Dashed lines correspond to the non-private training results depicted as upper bounds. Source data are provided as a Source Data file.

## Discussion

The main contribution of our paper is to analyse the impact of strong objective guarantees of privacy on the fairness enjoyed by specific patient subgroups in the context of AI model training on real-world medical datasets.

Across all levels of privacy protection, training with DP still yielded models exhibiting AUROC scores of 83% at the highest privacy level and 87% at an $\epsilon = 7.89$ on the UKA-CXR dataset. The fact that the model maintained a relatively high AUROC even at $\epsilon = 0.29$ is remarkable, and we are unaware of any prior work to report such a strong level of privacy protection at this level of model accuracy on clinical data. Our results thus exemplify that, through careful choice of architectures and best practices for the training of DP models, the use of model pretraining on a related public dataset, and the availability of sufficient data samples, privately trained models require only very small additional amounts of private information from the training dataset to achieve high diagnostic accuracy on the tasks at hand.

For the PDAC dataset, even though private models at $\epsilon = 8.0$ are not significantly inferior compared to non-private counterparts, the effect of the lower amount of training samples is observable at more restrictive privacy budgets. Especially at $\epsilon \leq 1.06$, the negative effect of private training on the discrimination of patients in certain age groups becomes noticeable. This underscores the requirement for larger training datasets, which the objective privacy guarantees of DP can enable through incentivizing data sharing.

Our analysis of the per-diagnosis performance of models that are trained with and without privacy guarantees shows that models discriminate against diagnoses that are underrepresented in the training set in both private and non-private training. This finding is not unusual

and several examples can be found in[51]. However, the drop in performance between private and non-private training is uncorrelated to the sample size. Instead, the difficulty of the diagnosis seems to drive the difference in AUROC between the two settings. Concretely, diagnostic performance under privacy constraints suffers the most for those classes, which already have the lowest AUROC in the non-private setting. Conversely, diagnoses that are predicted with the highest AUROC suffer the least when DP is introduced.

Previous works investigating the effect of DP on fairness show that privacy preservation amplifies discrimination[33]. This effect is limited to very low privacy budgets in our study. Our models remain fair despite at the levels of privacy protection typically used for training state-of-the-art models in current literature[25], likely due to our real-life datasets' large size and/or high quality.

The effects we observed are not limited to within-domain models. Indeed, in a concurrent work, we investigated the effects of DP training on the domain generalizability of diagnostic medical AI models[52]. Our findings revealed that even under extreme privacy conditions, DP-trained models show comparable performance to non-DP models in external domains.

Our analysis of fairness related to patient age showed that older patients are discriminated against both in the non-private and private settings. On UKA-CXR, age-related discrimination remains approximately constant with stronger privacy guarantees. On the other hand, young patients enjoy overall lower model discrimination in the non-private and the private setting. Interestingly, young patients seem to profit more from stronger privacy guarantees, as they enjoy progressively more fairness privilege with increasing privacy protection level.

This holds despite the fact that patients under 30 represent the smallest fraction of the UKA-CXR dataset. The privilege of young patients is most likely due to a confounding variable, namely the lower complexity of imaging findings in younger patients due to their improved ability to cooperate during radiograph acquisition, resulting in better discrimination of the pathological finding on a more homogeneous background (i.e., "cleaner") radiographs which are easier to diagnose overall[35,53] (see Fig. 6). This hypothesis should be validated in cohorts with a larger proportion of young patients, and we intend to expand on this finding in future work. On the PDAC dataset, classification accuracy remains approximately on par between age subgroups except at very restrictive privacy budgets, where older patients begin to suffer discrimination, likely due to the aforementioned imbalance between control and tumor cases and the overall smaller dataset coupled with a lack of pre-training. The analysis of model fairness related to patient sex for UKA-CXR shows that female patients (which – similar to young patients – are an underrepresented group) enjoy a slightly higher diagnostic accuracy than male patients for almost all privacy levels and vice versa on the PDAC dataset. However, effect size differences were found to be small, so that this finding can also be explained by variability between models or by the randomness in the training process. Further investigation is thus required to elucidate the aforementioned effects.

Furthermore, there is no final conclusion for which fairness measure is preferable. In our study we focused on the statistical parity difference, however, there are other works proposing other measures. One, which recently received attention, is the underdiagnosis rate of subgroups[54]. We evaluated this for the PDAC dataset and found that in principle it shows the same trends as the statistical parity difference (see Supplementary Tables 9 and 10).

In conclusion, we analyzed the usage of privacy-preserving neural network training and its implications on utility and fairness for a relevant diagnostic task on a large real-world dataset. We showed that the utilization of specialized architectures and targeted model pre-training allows for high model accuracy despite stringent privacy guarantees. This enables us to train expert-level diagnostic AI models even with privacy budgets as low as $\varepsilon < 1$, which – to our knowledge – has not been shown before, and represents an important step towards the widespread utilization of differentially private models in radiological diagnostic AI applications. Moreover, our findings that the introduction of differential privacy mechanisms to model training does – in most cases – not amplify unfair model bias regarding patient age, sex or comorbidity signifies that – at least in our use case – the resulting models abide by important non-discrimination principles of ethical AI. We are hopeful that our findings will encourage practitioners and clinicians to introduce advanced privacy-preserving techniques such as differential privacy when training diagnostic AI models.

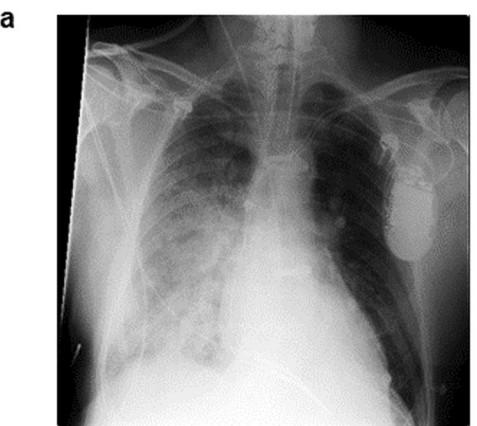
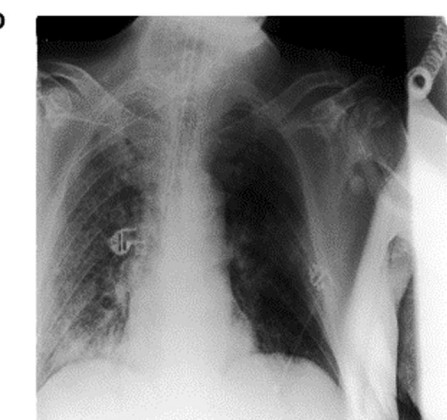
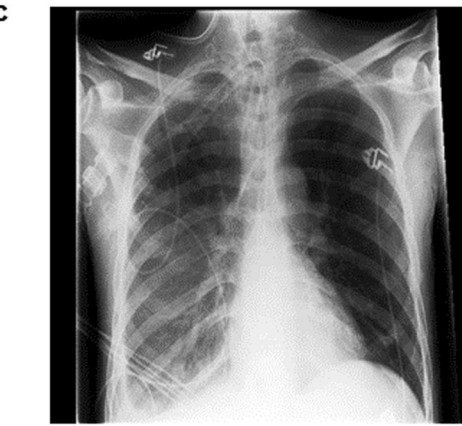
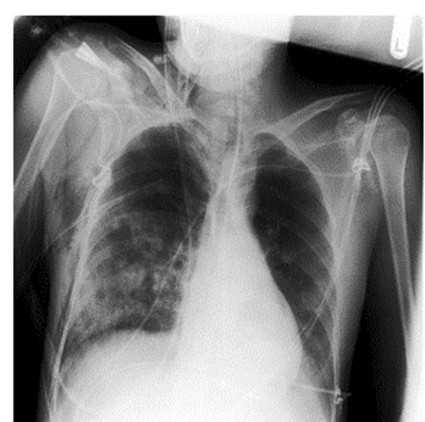
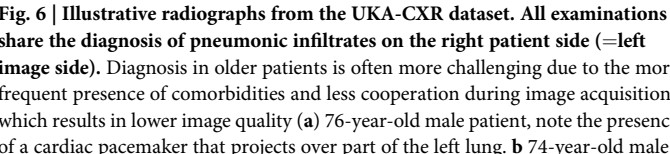

**Fig. 6 | Illustrative radiographs from the UKA-CXR dataset. All examinations share the diagnosis of pneumonic infiltrates on the right patient side (=left image side).** Diagnosis in older patients is often more challenging due to the more frequent presence of comorbidities and less cooperation during image acquisition which results in lower image quality (**a**) 76-year-old male patient, note the presence of a cardiac pacemaker that projects over part of the left lung. **b** 74-year-old male patient with challenging image acquisition: part of the lower right lung is not properly depicted. **c** 39-year-old male patient, the lungs are well inflated and pneumonic infiltrates can be discerned even though they are less severe. **d** 33-year-old male patient with challenging image acquisition, yet both lungs can be assessed (almost) completely.

## Data availability

The UKA-CXR dataset is not publicly accessible, in adherence to the policies for patient privacy protection at the University Hospital RWTH Aachen in Aachen, Germany. Similarly, the PDAC dataset cannot be publicly shared due to patient privacy considerations, as it is an in-house dataset at Klinikum Rechts der Isar, Munich, Germany. Data access for both datasets can be granted upon reasonable request to the corresponding author. Source data presented in Figures are available as Supplementary Data 1.

## Code availability

All source codes used for UKA-CXR for training and evaluation of the deep neural networks, differential privacy, data augmentation, image analysis, and preprocessing are publicly available at https://github.com/tayebiarasteh/DP_CXR. All code for the experiments was developed in Python 3.9 using the PyTorch 2.0 framework. The DP code was developed using Opacus 1.4.0[55]. Considering the utilization of equivalent computational resources, the time taken for the DP training to converge was approximately 10 times longer, in terms of total training time, than that required for the non-DP training with a similar network architecture. All code for the analyses on the PDAC dataset are available at https://github.com/TUM-AIMED/2.5DAttention. All source codes for both datasets are permanently archived on Zenodo and are accessible via[56] and[57].

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

## Acknowledgements

STA is funded and partially supported by the Radiological Cooperative Network (RACOON) under the German Federal Ministry of Education and Research (BMBF) grant number 01KX2021. AZ and GK were supported by the German Ministry of Education and Research (BMBF) under Grant Number 01ZZ2316C (PrivateAIM). DT is supported from the Deutsche Forschungsgemeinschaft (DFG) (TR 1700/7-1) as well as by the German Federal Ministry of Education (TRANSFORM LIVER, 031L0312A; SWAG, 01KD2215B) and the European Union's Horizon Europe and innovation programme (ODELIA [Open Consortium for Decentralized Medical Artificial Intelligence], 101057091). RB was supported by Deutsches Konsortium für Translationale Krebsforschung (DKTK). GK and DR have been funded by the German Federal Ministry of Education and Research and the Bavarian State Ministry for Science and the Arts through the Munich Centre for Machine Learning. DR was supported by ERC Grant Deep4MI (no. 884622). The funders played no role in the design or execution of the study. The authors of this work take full responsibility for its content.

## Author contributions

The formal analysis was conducted by S.T.A., A.Z., D.T. and G.K. The original draft was written by S.T.A. and A.Z. and edited by D.T. and G.K. The experiments as well as the software development for UKA-CXR were performed by S.T.A. and for PDAC by A.Z. Statistical analyses were performed by A.Z. and S.T.A. D.T. and G.K. provided clinical and technical expertise. S.T.A., A.Z., C.K., M.M., S.N., R.B., D.R., D.T. and G.K. read the manuscript and agreed to the submission of this paper.

## Funding

## Competing interests

The authors declare no competing interests.
