## [Peer Review File · Communications Medicine]

Reviewers' comments:

Reviewer #1 (Remarks to the Author):

The paper presents a study of the effect on performance, bias and fairness of DP techniques in a 2D radiological setting.

My first major conceptual concern with the paper regards the topic of privacy itself. Authors do not describe the attack mechanism or the setup where hypothetically being able to extract a non-perfect data sample from a dataset would be considered a privacy concern, or why. Currently, most (if not all) AI-based medical devices in use and approved by regulators have not been trained with DP or with privacy protections, and they are not deemed to be a privacy risk, so it appears that it is not a real-world concern. Also, most methods that can "attack" such models assume that membership or rough data reconstruction somehow breaches privacy, when the reality is that, for example, knowing that data from a person might have been used to train a model with a probability of X is not the kind of privacy concern that most would consider relevant or harmful. Note that DP and privacy protections are important in other environments (eg federated learning) where significant amounts of data can be extracted during training if models are left unprotected, but it is hard (in my view) to make a case for the need for DP in training if the model attack or privacy risk only occurs during inference.

A second conceptual concern that I have is the "privacy budget". The concept itself makes sense, of course, but I take issue with the arguments around it. Either we say that privacy is absolute (we want zero risk of re-identification) and thus almost nothing can be learned, or we accept that privacy is not absolute and thus have to balance risk/outcome. Without a good model of harm due to privacy concerns (how do we value the harm caused by a hypothetical re-identification), I cannot really see how to set an appropriate privacy budget. One could argue that model accuracy needs to be as good as possible (it would be unethical otherwise), and thus, assuming that DP harms model performance, there should be no privacy protection whatsoever.

From a methodological point of view, it would be good to understand when and if the findings of the paper can be applied to other models/loss functions. ResNet9, for example, is not a very complex/parameter-heavy model, and thus has limited overfitting/memorisation capabilities than much bigger or much smaller models. The same comment would apply to 3D networks, where the model_parameter/training_sample ratio is larger than for 2D models. If we are to make general statements about how DP can be used in healthcare radiology AI, then the methodology needs to be a bit more general.

The overall comparison between Epsilons for different findings and different groups is very well executed and comprehensive. However, no statistical comparison (e.g. statistical tests) is provided beyond first order statistics. For example, given that the std in Table 2 is so small, the difference between non-private and other epsilon values should be statistical significant, meaning that statements such as "a mere 2.6% performance decrease" become very strongly statistically significant. This leads me to my last point, which is related to statements surrounding accuracy. Statements in the paper, such as "such a strong level of privacy protection at this level of model accuracy on clinical data" are too strong and not ethically-framed. A drop from 0.9 to 0.83 ROCAUC is very clinically relevant; even 0.87 of Epsilon=7.89 is very much smaller than 0.9, and one would need to ethically justify why dropping 0.03 in performance is admissible for a small (but not full)

protection against reidentification.

Reviewer #2 (Remarks to the Author):

The paper aim to assess the effectiveness of differential privacy neural network training techniques when applied to a large-scale Chest Xray dataset. There are numerous results being presented which generally show that even for small values of ϵ (higher privacy guarantees), the trained model does not exhibit extreme loss in accuracy (AUROC gets slightly degraded). The paper is well written, clearly explaining the goals, techniques, and results.

Some of the conclusions are very interesting: e.g. the fact that the model performance is affected not on the underrepresented classes but more on the classes with inferior accuracy in the non-private case, or the fact that Xrays of younger people are more resilient against DP-training.

It is mentioned throughout the paper that this type of evaluation is novel for large-scale Xray datasets, however there seem to be several papers discussing this. Some of them should be included in the prior work:

- "Defending against Reconstruction Attacks through Differentially Private Federated Learning for Classification of Heterogeneous Chest X-Ray Data", Joceline Ziegler et al.
- "Deep learning-based patient re-identification is able to exploit the biometric nature of medical chest X-ray data". Kai Packhäuser et. al.
- "FedSGDCOVID: Federated SGD COVID-19 Detection under Local Differential Privacy Using Chest X-ray Images and Symptom Information". Trang-Thi Ho et al.

Other points for improving the manuscripts:

- The epsilon (ϵ) parameter should be better explained for readers not accustomed to differential privacy (a quick google search explains it well)
- It would be nice to see the same experiments when not using a pretrained model, as the pretraining was done in a non-private way. Would the same conclusions hold when training the model from scratch on the UKA-CXR dataset?

Reviewer #3 (Remarks to the Author):

The paper investigates differential privacy (DP) to train an AI model (ResNet9) on a large-scale in-house chest X-ray dataset for multi-class classification. The presentation is clear, and the methods are technically sound. The technical novelties of the work, however, are limited. The authors use readily available tools (PyTorch for deep learning & Opacus for DP) to run the experiments. At the same time, there is a novelty in the application and detailed analysis of the results, which makes the work interesting to the field.

My main concerns with this work are listed below

1. The performance drop in AUROC reported in the abstract still seems statistically significantly

worse. A statistical test should be performed to evaluate whether the performance drop is acceptable, i.e. does not impact the performance of the model significantly. I expect it would. The claim that "accuracy reductions to be negligible compared to non-private training" need to be justified by statistical analysis.

2. Furthermore, it is hard to interpret a practically useful privacy budget in the context of this image classification model. Ideally, the authors could use an attack such as membership inference to quantify how many individual patient images could be reconstructed from the trained model. Otherwise, the claim that it is important to accept the drop in performance (even if relatively small) is unjustified. See for example [1].

3. While the approach was evaluated on a large dataset, the task (2D chest x-ray classification) is rather limited. It is unclear if the findings generalize to other medical imaging modalities, in particular 3D models and model architectures, e.g., U-Net for 3D image segmentation. Other medical imaging tasks such as brain imaging might even exhibit higher privacy concerns as they have to potential to identify individuals by the structure of the brain.

4. Furthermore, it would be interesting to the ready to see how non-private and privately trained models compare performance on an unseen dataset. That could give further insights into the generalizability of models trained with DP and, therefore, their usefulness in practice. I expect that DP could even help with the generalizability of the model as it acts similarly to other regularization approaches and should avoid overfitting to particular characteristics of the training data.

Below are some comments to improve the work for clarity and for easier interpretation by the reader.

5. How do delta and l2 norm clipping influence the results?

6. Page 3, from my reading of Pati et al., does not employ DP. Are the authors sure this is the correct reference? Some more relevant works investigating DP in federated learning for medical images could [2,3].

7. Table 1 would be better to use graphical representations of data statistics, e.g., bar plots or pie charts.

8. Ethnicity (if available) could be another sub-group analysis of the impact of DP for this task.

9. What are the runtime/memory trade-offs of using DP? This should be compared.

10. In 2.3.1, explain why batch normalization is not compatible with DP.

11. The paper uses a fixed batch size of 128. What would be the influence of batch size on privacy preservation? Again, a quantitative method to judge the privacy of the trained model would be needed.

12. Similarly in section 3.1 Which epsilon value is considered private? "stringent privacy guarantees" is not a quantifiable term.

13. Table 2, how come the statistical variation for each class is 0.00? I wonder if the bootstrapping works as expected to evaluate the variation in performance.

14. Table 2, for better interpretation, I would suggest to represent this as a x/y plot with epsilon on x-axis and different colored lines for each class accuracy. Non-private can be shown as horizontal lines as upper bounds. This way Fig. 3 could be moved to supplemental.

15. Table 3, again, graphical data will be easier to interpret for the reader. I would suggest moving the table's raw data to supplemental.

16. Fig. 4 should show upper bounds of accuracy of the non-private model.

Refs:

[1] Yin, Hongxu, et al. "Dreaming to distill: Data-free knowledge transfer via deepinversion." Proceedings of the IEEE/CVF Conference on Computer Vision and Pattern Recognition. 2020.

[2] Li, Wenqi, et al. "Privacy-preserving federated brain tumour segmentation." Machine Learning in Medical Imaging: 10th International Workshop, MLMI 2019, Held in Conjunction with MICCAI 2019, Shenzhen, China, October 13, 2019, Proceedings 10. Springer International Publishing, 2019.

[3] Hatamizadeh, Ali, et al. "Do gradient inversion attacks make federated learning unsafe?." IEEE Transactions on Medical Imaging (2023).

Dear Editors and Reviewers:

We would like to thank the editors and reviewers for their time and constructive comments. Following their comments, we have substantially and carefully revised the entire manuscript. Most notably, in response to concerns by the reviewers we have added another dataset, and experiments with additional model architectures. As these were major changes, we had to adapt and rewrite some parts of the manuscript. Please find our response to the reviewer's remarks below along with their resulting changes to the manuscript. We invite the reviewers to also check the revised draft.

Reviewers' comments:

Reviewer #1 (Remarks to the Author):

The paper presents a study of the effect on performance, bias and fairness of DP techniques in a 2D radiological setting.

My first major conceptual concern with the paper regards the topic of privacy itself. Authors do not describe the attack mechanism or the setup where hypothetically being able to extract a non-perfect data sample from a dataset would be considered a privacy concern, or why. Currently, most (if not all) AI-based medical devices in use and approved by regulators have not been trained with DP or with privacy protections, and they are not deemed to be a privacy risk, so it appears that it is not a real-world concern. Also, most methods that can "attack" such models assume that membership or rough data reconstruction somehow breaches privacy, when the reality is that, for example, knowing that data from a person might have been used to train a model with a probability of X is not the kind of privacy concern that most would consider relevant or harmful. Note that DP and privacy protections are important in other environments (eg federated learning) where significant amounts of data can be extracted during training if models are left unprotected, but it is hard (in my view) to make a case for the need for DP in training if the model attack or privacy risk only occurs during inference.

Response: We thank the reviewer for bringing up this interesting aspect of this topic, and would like to offer a slightly different perspective on this point. Indeed, most medical AI systems have been trained without formal privacy protections, however we would contend that this does not mean that the issue is not a real-world concern, just that training these models without formal protections does not constitute a legal privacy violation. However, privacy legislation is not necessarily up to date with current state-of-the art attacks. Therefore, if legislation changes, these models will potentially be deemed non-conformant and lose their certification. The benefit of a formal privacy guarantee like DP is that it certifies conformity to a specific notion of privacy protection *independent* of any future change in legislation. We would also like to point out that the kind of reconstruction attacks possible against machine learning models trained without DP are of increasingly high fidelity, such that it is likely that the opinion of patients or regulators could also change in light of these new attacks. Thus, while current workflows may not (yet) appear to benefit from formal privacy guarantees, this could change in the near future, and a future-proof guarantee like DP is useful in such a circumstance. The

reviewer is correct that federated learning is a domain in which DP is already very useful. We have expanded the manuscript to incorporate the points above. The corresponding section now reads:

"Most, if not all, currently deployed machine learning models are trained without any formal privacy-preservation technique. It is especially crucial to employ such techniques in federated scenarios, where much more granular information about the training process can be extracted, or even the training process itself can be manipulated by a malicious participant [7,8]. Moreover, trained models can be attacked to extract training data through so-called model inversion attacks [9,10,11]. We also note that such attacks work better if the models have been trained on less data, which is especially concerning since even most FDA-approved AI algorithms have been trained on fewer than 1000 cases [12]. Creating a one-to-one correspondence between a successful attack and the resulting "privacy risk" requires a case-by-case consideration. The legal opinion (e.g., the GDPR) seems to have converged on the notion of singling out/ re-identification. Even from the aspect of newer legal frameworks, such as the EU AI act, which demand "risk moderation" rather than directly specifying "privacy requirements", DP can be seen as the optimal tool as it can quantitatively bound both the risk of membership inference (MI) [13,14] and data reconstruction [15]. Moreover, this was also shown empirically for both aforementioned attack classes [16,17,18,19]. It is also known that DP, contrary to de-identification procedures such as k -anonymity, provably protects against the notion of singling out [20,21]."

A second conceptual concern that I have is the "privacy budget". The concept itself makes sense, of course, but I take issue with the arguments around it. Either we say that privacy is absolute (we want zero risk of re-identification) and thus almost nothing can be learned, or we accept that privacy is not absolute and thus have to balance risk/outcome. Without a good model of harm due to privacy concerns (how do we value the harm caused by a hypothetical re-identification), I cannot really see how to set an appropriate privacy budget. One could argue that model accuracy needs to be as good as possible (it would be unethical otherwise), and thus, assuming that DP harms model performance, there should be no privacy protection whatsoever.

Response: We thank the reviewer again for discussing the big picture. This discussion touches on sociological, ethical, and political aspects and can thus not be conclusively answered in a technical study like ours. The reviewer is correct that absolute privacy can only come at the price of being able to learn almost nothing. Moreover, the reviewer correctly touches on a common "pain point" of DP, namely appropriately setting the privacy budget. We would however argue that this does not represent a problem with DP. To the contrary, non-quantitative privacy techniques which have no budget also tend to fail catastrophically, which we would contend is equally problematic. Regarding the topic of whether accuracy or privacy is the greater good, this is a debate with a profound socio-ethical flavour. If there is a consensus that accuracy trumps privacy in every instance, not using any privacy protection is a legitimate strategy, but this is not the case in all legislations or societies and thus, the trade-off is essentially mandated. (The same goes with many other trade-offs between personal freedoms and benefits and those of society as a whole). This is however not the point we are making in the paper. Instead, we merely try to relate formal privacy techniques to the model fairness of underrepresented groups.

Our personal stance on the general topic of privacy-utility trade-offs is that (1) if the utility trade-off is minimal, it is definitely preferable to use formal privacy protection methods, and (2) it might even be the key to higher utility, as the bottleneck for all medical machine learning algorithms so far is the availability of large and diverse datasets. Stronger incentives towards data contributors through the utilisation of privacy technology could lead to such datasets becoming available for the training of ML algorithms.

We have added the following discussion to our manuscript related to the aforementioned points:

"We note that absolute privacy (i.e. zero risk) is only possible if no information is present [24]. This is, for example, the case in encryption methods, which are perfectly private as long as data is not decrypted. Note that training models e.g. via homomorphic encryption does, however, not offer such perfect privacy guarantees, as the information learned by the model is actually revealed at inference time through the model's predictions. Thus, without the protection of differential privacy, no formal barrier stands between the sensitive data and an attacker (beyond potential imperfections of the attack algorithm, which are usually not controllable *a priori*). DP offers the ability to upper-bound the risk of successful privacy attacks while still being able to draw conclusions from the data. Determining the exact privacy budget is challenging, as it is a matter of *policy*. The technical perspective can provide insight into the appropriate budget level, as it is possible to quantify the risk of a successful attack at a given privacy budget compared to the model utility that can be achieved. The trade-offs between model utility and privacy preservation are also a matter of ethical, societal and political debate."

[...]

"In this work, we aim to elucidate the connection between using formal privacy techniques and the fairness towards underrepresented groups in the sensitive setting of medical use-cases."

From a methodological point of view, it would be good to understand when and if the findings of the paper can be applied to other models/loss functions. ResNet9, for example, is not a very complex/parameter-heavy model, and thus has limited overfitting/memorisation capabilities than much bigger or much smaller models. The same comment would apply to 3D networks, where the model_parameter/training_sample ratio is larger than for 2D models. If we are to make general statements about how DP can be used in healthcare radiology AI, then the methodology needs to be a bit more general.

Response: We thank the reviewer for this suggestion, with which we agree. We selected the ResNet9 architecture for its efficiency in diagnosing chest radiographs with precision. This approach enabled us to train various DP networks (with different privacy budgets), allowing us to focus our analyses on utility and fairness.

We have now expanded our study by conducting further experiments on more complex architectures compared to ResNet9. Specifically, we trained the EfficientNet B0 [1], DenseNet121 [2], and ResNet18 [3] architectures. A newly-created figure displays the results of DP training at various

epsilon values for these networks, compared with non-DP training. The trends observed were consistent with those for the ResNet9 architecture.

Furthermore, we added a second dataset containing 1,625 abdominal 3D CT scans, where we classified whether patients have a pancreatic ductal adenocarcinoma. We find that again, subgroups that are better predictable in the standard setting also have smaller impacts on the utility when trained privately.

Furthermore, to prevent domain-specific bias in our results, we expanded our study beyond the radiological use-case with the Artificial Intelligence for Robust Glaucoma Screening (AIROGS) dataset [4]. This dataset comprises 101,354 ocular fundus images from approximately 60,000 patients of diverse ethnicities, aimed at detecting the presence of referable glaucoma, where we observed a trend that aligns with our observations for chest radiographs.

Accordingly, we have incorporated the following paragraphs into our manuscript:

"Additional remarks on privacy-utility trade-off

Varying model architectures

In addition to the ResNet9 architecture reported in the main manuscript, we additionally used three more architectures: An EfficientNet B0, with 4,017,796 parameters, adhering to the original implementation proposed by Tan et al. [56], with the sole exception of replacing all batch normalization layers with group normalization; DenseNet121, with 6,962,056 parameters, following the original design put forth by Huang et al. [57], again with the exclusive modification of substituting batch normalization layers with group normalization; and ResNet18, with 11,180,616 parameters, following the original blueprint developed by He et al. [40], with the unique alteration of replacing batch normalization layers with group normalization. All three models displayed a trend consistent with the utility penalties we observed for ResNet9 in both DP and non-DP training. Compare also Figure 10.

Further datasets

To prevent domain-specific bias in our results, we employed the Artificial Intelligence for Robust Glaucoma Screening (AIROGS) dataset [55]. This dataset comprises 101,354 RGB ocular fundus images from approximately 60,000 patients of diverse ethnicities, aimed at detecting the presence of referable glaucoma. We allocated 80% of the patients—both with and without glaucoma—to the training set, reserving the remaining 20% for the test set. Image pre-processing involved cropping and other schemes as detailed in [58] and [59]. The images were resized to a dimension of $3 \times 224 \times 224$, with 3 representing the number of channels. We adopted the same EfficientNet B0 network architecture, with identical DP and non-DP training parameters as described earlier, with the same $\delta = 6 \times 10^{-6}$. The network was pre-trained on the ImageNet [60] dataset. Figure 16 shows a similar trend as our observations on chest radiographs regarding the privacy-utility trade-off."

[1] Tan, Mingxing, and Quoc Le. "Efficientnet: Rethinking model scaling for convolutional neural networks." International conference on machine learning. PMLR, 2019.

- [2] Huang, Gao, et al. "Densely connected convolutional networks." Proceedings of the IEEE conference on computer vision and pattern recognition. 2017.
- [3] He, Kaiming, et al. "Deep residual learning for image recognition." Proceedings of the IEEE conference on computer vision and pattern recognition. 2016.
- [4] de Vente, Coen, et al. "AIROGS: Artificial Intelligence for RObust Glaucoma Screening Challenge." arXiv preprint arXiv:2302.01738 (2023).

The overall comparison between Epsilons for different findings and different groups is very well executed and comprehensive. However, no statistical comparison (e.g. statistical tests) is provided beyond first order statistics. For example, given that the std in Table 2 is so small, the difference between non-private and other epsilon values should be statistically significant, meaning that statements such as "a mere 2.6% performance decrease" become very strongly statistically significant. This leads me to my last point, which is related to statements surrounding accuracy. Statements in the paper, such as "such a strong level of privacy protection at this level of model accuracy on clinical data" are too strong and not ethically-framed. A drop from 0.9 to 0.83 ROCAUC is very clinically relevant; even 0.87 of Epsilon=7.89 is very much smaller than 0.9, and one would need to ethically justify why dropping 0.03 in performance is admissible for a small (but not full) protection against reidentification.

Response: We thank the reviewer for their remark. We would again like to point out that our paper does not aim to provide a definite answer whether and at what level of privacy is called for in medical machine learning. Our aim is to elucidate the relationship of privacy, utility, and fairness in an exemplary setting to provide a basis for the public as well as ethicists to discuss general guidelines for practitioners.

We agree that our description was not based on a quantifiable measure. We described it as such as most practitioners would likely find a model with 90% compared to 87.4% as performing almost equally well. We have now removed the corresponding sentence from the abstract. Furthermore, we changed the following sentence in the results section: "Training with DP decreases all results slightly yet significantly (Hanley & McNeil-test p -value < 0.001)." To avoid any such inaccuracies in the remaining manuscript we moreover changed the contributions as Reviewer #3 suggested (see below) and in the discussion the sentence *Across all levels of privacy protection, training with DP only resulted in mild AUROC reductions to [...]* "training with DP still yielded models exhibiting AUROC scores of 83% at the highest privacy level and 87% at an $\epsilon=7.89$."

Reviewer #2 (Remarks to the Author):

The paper aim to assess the effectiveness of differential privacy neural network training techniques when applied to a large-scale Chest Xray dataset. There are numerous results being presented

which generally show that even for small values of ϵ (higher privacy guarantees), the trained model does not exhibit extreme loss in accuracy (AUROC gets slightly degraded). The paper is well written, clearly explaining the goals, techniques, and results.

Some of the conclusions are very interesting: e.g. the fact that the model performance is affected not on the underrepresented classes but more on the classes with inferior accuracy in the non-private case, or the fact that Xrays of younger people are more resilient against DP-training.

It is mentioned throughout the paper that this type of evaluation is novel for large-scale Xray datasets, however there seem to be several papers discussing this. Some of them should be included in the prior work:

- "Defending against Reconstruction Attacks through Differentially Private Federated Learning for Classification of Heterogeneous Chest X-Ray Data", Joceline Ziegler et al.
- "Deep learning-based patient re-identification is able to exploit the biometric nature of medical chest X-ray data". Kai Packhäuser et. al.
- "FedSGDCOVID: Federated SGD COVID-19 Detection under Local Differential Privacy Using Chest X-ray Images and Symptom Information". Trang-Thi Ho et al.

Response: We thank the reviewer for this remark. We are not claiming to be the first to evaluate the impact of DP on chest X-ray classification, but the first to study the trade-offs between privacy, utility, and fairness on real-world medical datasets. Following this comment and the remarks of other reviewers, we substantially expanded our prior works section and included the first two listed works. We have decided against listing the third reference, as they study much lower privacy guarantees ($\epsilon > 49$) in a federated setup and are thus hardly comparable to our work. The changes read now as follows:

"The need for the use of differential privacy (DP) has been illustrated by Packhäuser et al. [31], who showed that it is trivial to match chest x-rays of the same patient, which directly enables re-identification attacks; this was similarly shown in tabular databases by Narayanan et al. [32] [...] Li et al. [33] investigated privacy-utility trade-offs in the combination of advanced federated learning schemes and DP methods on a brain tumor segmentation dataset. They find that DP introduces a considerable reduction in model accuracy in the given setting.

Hatamizadeh et al. [23] illustrated that the use of federated learning alone is unsafe, as training data can be reconstructed in such settings. Ziegler et al. [34] reported similar findings when evaluating privacy-utility trade-offs for a chest x-ray classification on a public dataset."

Other points for improving the manuscripts:

- The epsilon (ϵ) parameter should be better explained for readers not accustomed to differential privacy (a quick google search explains it well)

Response: We thank the reviewer for this comment, and agree that DP is not yet well-known enough in the medical imaging community to omit an accessible explanation. Hence we added the following to the introduction of DP:

"In more intuitive terms, DP is a guarantee given from a data processor to a data owner that the risks of adverse events which can occur due to the inclusion of their data in a database are not significantly increased compared to the risks of such events when their data is not included. The parameters ϵ and δ together form what is typically called a *privacy budget*. Higher values of ϵ and δ correspond to a looser privacy guarantee and *vice versa*. With some terminological laxity, ϵ can be considered a measure of the *privacy loss* incurred, whereas δ represents a (small) probability that this privacy loss is exceeded."

- It would be nice to see the same experiments when not using a pretrained model, as the pretraining was done in a non-private way. Would the same conclusions hold when training the model from scratch on the UKA-CXR dataset?

Response: We appreciate the reviewer's suggestion. In response, we have conducted additional experiments which we believe further enrich our manuscript. These results are depicted in a newly incorporated figure. Consequently, we have updated our Results section:

"Moreover, for UKA-CXR, the use of pre-training helps to boost model performance and reduce the amount of additional information the model needs to learn 'from scratch' and consequently reduces the privacy budgets required (refer to Supplementary Figure 9)"

Reviewer #3 (Remarks to the Author):

The paper investigates differential privacy (DP) to train an AI model (ResNet9) on a large-scale in-house chest X-ray dataset for multi-class classification. The presentation is clear, and the methods are technically sound. The technical novelties of the work, however, are limited. The authors use readily available tools (PyTorch for deep learning & Opacus for DP) to run the experiments. At the same time, there is a novelty in the application and detailed analysis of the results, which makes the work interesting to the field.

My main concerns with this work are listed below

1. The performance drop in AUROC reported in the abstract still seems statistically significantly worse. A statistical test should be performed to evaluate whether the performance drop is acceptable, i.e. does not impact the performance of the model significantly. I expect it would. The claim that "accuracy reductions to be negligible compared to non-private training" need to be justified by statistical analysis.

Response: We thank the reviewer for this remark. Indeed the reviewer's intuition is correct that all non-privately trained models statistically significantly outperform the private algorithms. We acknowledge that negligible in this context is based on our perception. To avoid any such confusion we changed the sentence as follows: **We reach 97% of the non-private AUROC on the UKA-CXR dataset through the utilization of transfer learning on public datasets and careful choice of architecture.** Following this and the last comment by Reviewer #1 we also changed the abstract and the discussion (see above).

2. Furthermore, it is hard to interpret a practically useful privacy budget in the context of this image classification model. Ideally, the authors could use an attack such as membership inference to quantify how many individual patient images could be reconstructed from the trained model. Otherwise, the claim that it is important to accept the drop in performance (even if relatively small) is unjustified. See for example [1].

Response: We thank the reviewer for this comment. Indeed the empirical evaluation of DP via reconstruction attacks is a very intuitive way of highlighting the benefits of such training. This has been investigated several times before. Hence, following this and a comment by Reviewer #1 we added the following sentence:

"Even from the aspect of newer legal frameworks, such as the EU AI act, which demand "risk moderation" rather than directly specifying "privacy requirements", DP can be seen as the optimal tool as it can quantitatively bound both the risk of membership inference (MI) [13,14] and data reconstruction [15]. Moreover, this was also shown empirically for both aforementioned attack classes [16,17,18,19]."

3. While the approach was evaluated on a large dataset, the task (2D chest x-ray classification) is rather limited. It is unclear if the findings generalize to other medical imaging modalities, in particular 3D models and model architectures, e.g., U-Net for 3D image segmentation. Other medical imaging tasks such as brain imaging might even exhibit higher privacy concerns as they have the potential to identify individuals by the structure of the brain.

Response: We thank the reviewer for this suggestion, with which we agree. We selected the ResNet9 architecture for its efficiency in diagnosing chest radiographs with precision. This approach enabled us to train various DP networks (with different privacy budgets), allowing us to focus our analyses on utility and fairness.

We have expanded our study by conducting further experiments on more complex architectures compared to ResNet9. Specifically, we trained the EfficientNet B0 [1], DenseNet121 [2], and ResNet18 [3] architectures. A newly-created figure displays the results of DP training at various epsilon values for these networks, compared with non-DP training. The trends observed were consistent with those for the ResNet9 architecture.

Furthermore, we extended our manuscript and analysed a second dataset in regards to fairness of subgroups. For this we added a binary classification task on an abdominal 3D CT dataset, where we classified whether patients suffer from a pancreatic ductal adenocarcinoma (PDAC). As this was a major addition, we have reworked our manuscript extensively. We find that this dataset supports our observations on the UKA-CXR dataset, that classes which are easier to predict in the standard setting have smaller impacts on the utility when trained privately.

Also, to prevent domain-specific bias in our results, we investigated another medical modality and employed the Artificial Intelligence for Robust Glaucoma Screening (AIROGS) dataset [4]. This dataset comprises 101,354 RGB ocular fundus images from approximately 60,000 patients of diverse ethnicities, aimed at detecting the presence of referable glaucoma, where we observed a trend that aligns with our observations for chest radiographs.

We have added this information to the Appendices section:

“Additional remarks on privacy-utility trade-off

Varying model architectures

In addition to the ResNet9 architecture reported in the main manuscript, we additionally used three more architectures: An EfficientNet B0, with 4,017,796 parameters, adhering to the original implementation proposed by Tan et al. [56], with the sole exception of replacing all batch normalization layers with group normalization; DenseNet121, with 6,962,056 parameters, following the original design put forth by Huang et al. [57], again with the exclusive modification of substituting batch normalization layers with group normalization; and ResNet18, with 11,180,616 parameters, following the original blueprint developed by He et al. [40], with the unique alteration of replacing batch normalization layers with group normalization. All three models displayed a trend consistent with the utility penalties we observed for ResNet9 in both DP and non-DP training. Compare also Figure 10.

Further datasets

To prevent domain-specific bias in our results, we employed the Artificial Intelligence for Robust Glaucoma Screening (AIROGS) dataset [55]. This dataset comprises 101,354 RGB ocular fundus images from approximately 60 000 patients of diverse ethnicities, aimed at detecting the presence of referable glaucoma. We allocated 80% of the patients—both with and without glaucoma—to the training set, reserving the remaining 20% for the test set. Image pre-processing involved cropping and other schemes as detailed in [58] and [59]. The images were resized to a dimension of $3 \times 224 \times 224$, with 3 representing the number of channels. We adopted the same EfficientNet B0 network architecture, with identical DP and non-DP training parameters as described earlier, with the same $\delta = 6 \times 10^{-6}$. The network was pre-trained on the ImageNet [60] dataset. Figure 16 shows a similar trend as our observations on chest radiographs regarding the privacy-utility trade-off.”

[1] Tan, Mingxing, and Quoc Le. "Efficientnet: Rethinking model scaling for convolutional neural networks." International conference on machine learning. PMLR, 2019.

- [2] Huang, Gao, et al. "Densely connected convolutional networks." Proceedings of the IEEE conference on computer vision and pattern recognition. 2017.
- [3] He, Kaiming, et al. "Deep residual learning for image recognition." Proceedings of the IEEE conference on computer vision and pattern recognition. 2016.
- [4] de Vente, Coen, et al. "AIROGS: Artificial Intelligence for RObust Glaucoma Screening Challenge." arXiv preprint arXiv:2302.01738 (2023).

4. Furthermore, it would be interesting to the ready to see how non-private and privately trained models compare performance on an unseen dataset. That could give further insights into the generalizability of models trained with DP and, therefore, their usefulness in practice. I expect that DP could even help with the generalizability of the model as it acts similarly to other regularization approaches and should avoid overfitting to particular characteristics of the training data.

Response: We thank the reviewer for this suggestion. In line with this, we have conducted a parallel study investigating the effects of DP training on the domain generalizability of diagnostic medical AI models. Our findings reveal that even under extreme privacy conditions ($\epsilon \approx 1$), DP-trained models show comparable performance to non-DP models in external domains. These findings are detailed in our pre-print study [1]. We incorporated this in the discussion section as follows:

"The effects we observed are not limited to within-domain models. Indeed, in a concurrent work, we investigated the effects of DP training on the domain generalizability of diagnostic medical AI models [51]. Our findings revealed that even under extreme privacy conditions, DP-trained models show comparable performance to non-DP models in external domains."

[1] Tayebi Arasteh, Soroosh, et al. "Preserving privacy in domain transfer of medical AI models comes at no performance costs: The integral role of differential privacy." arXiv preprint arXiv:2306.06503 (2023).

Below are some comments to improve the work for clarity and for easier interpretation by the reader.

5. How do delta and l2 norm clipping influence the results?

Response: We thank the reviewer for this comment, this might indeed not be entirely clear. Delta is part of the privacy budget. Hence, a higher value for delta would correspond to a looser privacy bound and thus likely lead to better model trainings. The L2 clipping norm is an additional hyperparameter like for example the learning rate. We have modified parts of the introduction as follows to clarify this:

"[...] The parameters ϵ and δ together form what is typically called a *privacy budget*. Higher values of ϵ and δ correspond to a looser privacy guarantee and *vice versa*. With some terminological laxity, ϵ can be considered a measure of the *privacy loss* incurred, whereas δ represents a (small) probability

that this privacy loss is exceeded. For deep learning workflows, δ is set to around the inverse of the database size.

[...] they have been clipped in L2-norm to ensure that their magnitude is bounded [7], where the clipping threshold is an additional hyperparameter in the training process (see Figure 1).“

By this we hope to clarify that delta is not directly a hyperparameter, but crucial to express the privacy budget.

6. Page 3, from my reading of Pati et al., does not employ DP. Are the authors sure this is the correct reference? Some more relevant works investigating DP in federated learning for medical images could [2,3].

Response: The reviewer is indeed right, we have mistakenly listed Pati as an example of DP training. We follow the recommendations of the reviewer and include their suggested publications as related work. It now reads as:

“Li et al. [33] investigated privacy-utility trade-offs in the combination of advanced federated learning schemes and DP methods on a brain tumour segmentation dataset. They find that DP introduces a considerable drop in model performance in the given setting. Hatamizadeh et al. [23] illustrated that the use of federated learning alone is unsafe, as training data can be reconstructed in such settings.”

7. Table 1 would be better to use graphical representations of data statistics, e.g., bar plots or pie charts.

Response: We thank the reviewer for the suggestion. We have included the suggested figure in the appendix. Furthermore, due to the inclusion of a second dataset we've split up this table and merged the most important numbers to the results table and moved the entire table to the Appendix.

8. Ethnicity (if available) could be another sub-group analysis of the impact of DP for this task.

Response: We value the reviewer's suggestion regarding the inclusion of ethnicity information. Regrettably, ethnicity data is not available for the cohorts used in our study.

9. What are the runtime/memory trade-offs of using DP? This should be compared.

Response: We thank the reviewer for this comment regarding the runtime/memory trade-offs of using DP. In response to the inquiry, our manuscript details an implementation using Opacus in PyTorch. Thus, we added the following sentence to the “Code and Data Availability” subsection of the

manuscript: "Considering the utilization of equivalent computational resources, the time taken for the DP training to converge is approximately 10 times longer, in terms of total training time, than that required for the non-DP training with a similar network architecture."

10. In 2.3.1, explain why batch normalization is not compatible with DP.

Response: We thank the reviewer for this remark, this might indeed not be clear to the general audience. Hence we added the following sentence to the manuscript:

"Batch Normalization is incompatible with DP-SGD, as per-sample gradients are required and batch normalization inherently intermixes information of all images in one batch."

11. The paper uses a fixed batch size of 128. What would be the influence of batch size on privacy preservation? Again, a quantitative method to judge the privacy of the trained model would be needed.

Response: We thank the reviewer for this comment. Indeed, the batch size has an influence on the privacy budget at a constant noise level as the sampling rate changes. However, it is possible to modify the noise level accordingly so that for varying batch sizes the same theoretical guarantees hold true, which would be appropriate for experiments with varying batch sizes. This is the technique we also used and thus batch size is independent of the privacy guarantees. To further underline this we added the following sentence to the manuscript:

"The noise multiplier was calculated such that for a given number of training steps, batch size, and maximum gradient norm, the privacy budget was reached in the last training step."

12. Similarly in section 3.1 Which epsilon value is considered private? "stringent privacy guarantees" is not a quantifiable term.

Response: The reviewer is right that stringent is not a quantitative term but a qualitative one. We use stringent, as our budgets are below many SOTA works, which typically use privacy budgets of up to 8 or larger (compare [1], [2]), while we denote a range from 0.29 to 7.89.

[1] De, Soham, et al. "Unlocking high-accuracy differentially private image classification through scale." (2022).

[2] Kurakin, Alexey, et al. "Toward training at imagenet scale with differential privacy." (2022).

13. Table 2, how come the statistical variation for each class is 0.00? I wonder if the bootstrapping works as expected to evaluate the variation in performance.

Response: We thank the reviewer for their remark. When employing bootstrapping we found the standard deviation of the AUC to fall within the range of 0.001 to 0.005 and to be consistent in the number of digits, we rounded to 0.00. We double checked and reckon that this low variation is due to the very large test set (n=39,809) that we are using.

14. Table 2, for better interpretation, I would suggest to represent this as a x/y plot with epsilon on x-axis and different colored lines for each class accuracy. Non-private can be shown as horizontal lines as upper bounds. This way Fig. 3 could be moved to supplemental.

Response: We thank the reviewer for their suggestion. We have transformed Table 2 from the original manuscript into a figure, in which the non-private model's results are depicted as upper bounds. Also, we have moved the original Fig. 3 to the supplementary section of the manuscript.

15. Table 3, again, graphical data will be easier to interpret for the reader. I would suggest moving the table's raw data to supplemental.

Response: We thank the reviewer for their feedback. We have converted the AUROC values from Table 3 of the original manuscript (as well as accuracy, specificity, and sensitivity values from supplemental tables) into a figure. This new figure also features the results of the non-private model as upper bounds.

16. Fig. 4 should show upper bounds of accuracy of the non-private model.

Response: We thank the reviewer for their suggestion. We have now incorporated the recommended change in the corresponding figure, where the results of the non-private model are depicted as the upper bound. Further, we have also made analogous modifications to Fig. 2 from the initial submission, now illustrating the non-private model's results as the upper bound.

Refs:

[1] Yin, Hongxu, et al. "Dreaming to distill: Data-free knowledge transfer via deepinversion." Proceedings of the IEEE/CVF Conference on Computer Vision and Pattern Recognition. 2020.

[2] Li, Wenqi, et al. "Privacy-preserving federated brain tumour segmentation." Machine Learning in Medical Imaging: 10th International Workshop, MLMI 2019, Held in Conjunction with MICCAI 2019, Shenzhen, China, October 13, 2019, Proceedings 10. Springer International Publishing, 2019.

[3] Hatamizadeh, Ali, et al. "Do gradient inversion attacks make federated learning unsafe?." IEEE Transactions on Medical Imaging (2023).

REVIEWERS' COMMENTS:

Reviewer #2 (Remarks to the Author):

None

Reviewer #3 (Remarks to the Author):

The authors have provided a substantial revision, and my comments have been sufficiently addressed. Analyzing differential privacy in additional datasets and imaging modalities is very valuable.

The work is an important step towards understanding the use of differentially private model training and can be an important reference for building clinically useful AI models.

The new statement in the prior work section of "Hatamizadeh et al [23] illustrated that the use of federated learning alone is unsafe, as training data can be reconstructed in such settings." is a bit misleading. Hatamizadeh et al. actually show that current reconstruction attacks can only work in very contrived FL scenarios such in which gradients are sent from individual image batches. It should be rephrased to "Hatamizadeh et al [23] illustrated that the use of federated learning alone can be unsafe in certain settings, ..."

I read through the responses to Reviewer 1 and believe the authors did a great job in refuting some of the concerns and addressing the issues raised. The revision is sufficient for acceptance in my point of view.

Dear Editors and Reviewers:

The authors would like to thank the editors and reviewers for their time and constructive comments. We remain enthusiastic about our work and appreciate the opportunity to respond. We have addressed all the concerns and strongly believe this process has made our research stronger. We hope that the article is now suitable for publication in *Communications Medicine*. Please see the individual responses below for each reviewer.

REVIEWERS' COMMENTS:

Reviewer #2:

None

Reviewer #3 (Remarks to the Author):

The authors have provided a substantial revision, and my comments have been sufficiently addressed. Analyzing differential privacy in additional datasets and imaging modalities is very valuable.

The work is an important step towards understanding the use of differentially private model training and can be an important reference for building clinically useful AI models.

The new statement in the prior work section of "Hatamizadeh et al [23] illustrated that the use of federated learning alone is unsafe, as training data can be reconstructed in such settings." is a bit misleading. Hatamizadeh et al. actually show that current reconstruction attacks can only work in very contrived FL scenarios such in which gradients are sent from individual image batches. It should be rephrased to "Hatamizadeh et al [23] illustrated that the use of federated learning alone can be unsafe in certain settings, ..."

I read through the responses to Reviewer 1 and believe the authors did a great job in refuting some of the concerns and addressing the issues raised. The revision is sufficient for acceptance in my point of view.

Response: We thank the reviewer for their positive assessment of our revisions and the acknowledgement of our work's significance. We have amended the statement about Hatamizadeh et al. [23] in our manuscript as suggested, to accurately convey their findings on the potential risks in certain federated learning scenarios. We are grateful for the reviewer's approval of our responses to Reviewer 1 and are pleased that our manuscript is considered ready for acceptance.